# TAMing Gliomas: Unraveling the Roles of Iba1 and CD163 in Glioblastoma

**DOI:** 10.3390/cancers17091457

**Published:** 2025-04-26

**Authors:** Haneya Fuse, Yuqi Zheng, Islam Alzoubi, Manuel B. Graeber

**Affiliations:** 1School of Medicine, Sydney Campus, University of Notre Dame, 160 Oxford Street, Sydney, NSW 2010, Australia; haneya.fuse@my.nd.edu.au; 2Ken Parker Brain Tumor Research Laboratories, Brain and Mind Centre, Faculty of Medicine and Health, University of Sydney, Sydney, NSW 2050, Australia; yzhe8012@gmail.com; 3School of Computer Science, The University of Sydney, J12/1 Cleveland St, Sydney, NSW 2008, Australia; ialz9547@uni.sydney.edu.au; 4University of Sydney Association of Professors (USAP), University of Sydney, Sydney, NSW 2006, Australia

**Keywords:** CD163, glioblastoma, Iba1, immunosuppression, macrophage infiltration, microglial activation, TAMs-glioma interaction, tumor-associated macrophages (TAMs), tumor microenvironment (TME)

## Abstract

Most gliomas are aggressive brain tumors that diffusely infiltrate surrounding brain tissue, making them difficult to treat. Our research focuses on tumor-associated macrophages (TAMs), which play a significant role in glioma growth. We are particularly interested in microglia, the resident brain macrophage precursors that normally express Iba1 but not CD163. In the early stages of glioma development, microglia are the primary source of TAMs. However, macrophages from external sources, such as blood and perivascular spaces, which typically produce CD163, contribute to the TAM cell pool, especially in necrobiotic tumor tissue. By studying the regulation of Iba1 and CD163, we aim to better understand the role of TAMs in gliomas.

## 1. Introduction

In 2024, Siegel et al. [1] estimated that 18,760 deaths from brain and other nervous system cancers would occur in the United States alone in that year. Glioblastoma (GBM) accounts for nearly half of all malignant brain tumors, with a median survival of just 14.6 months [2]. Furthermore, the mortality rate for GBM has remained largely unchanged since 1975 [1], highlighting the persistent lack of significant therapeutic advances despite decades of medical research. In recent years, the emergence of targeted therapies—including immune checkpoint inhibitors and chimeric antigen receptor T-cell (CAR-T) therapy—has markedly improved outcomes in several cancer types [3]. However, such innovative approaches have shown limited efficacy in GBM so far, and the current gold standard for treatment still relies on maximal safe surgical resection followed by concurrent radiotherapy and temozolomide chemotherapy [4].

Molecular profiling studies are providing new insights into potential therapies. For instance, a study published by Liu and Tang [5] reported the discovery of 127 mitogen-activated protein kinase (MAPK) genes upregulated in glioma, allowing for a potential C1/C2 classification of glioma subtypes with distinct prognostic implications. Similarly, voltage-gated sodium channel β3 subunit (SCN3B) has been identified as a prognostic biomarker for gliomas, specifically oligodendroglioma [6], where higher levels of SCN3B correlate with longer survival. CDK2, a key cell cycle regulator, is associated with GBM prognosis. Furthermore, lower CDK2 levels indicate a higher response to immunotherapy [7], suggesting its potential role as a biomarker and a therapeutic target. These findings suggest new avenues for personalized and targeted therapies in GBM.

GBM is a highly malignant brain tumor with a complex tumor microenvironment (TME) that harbors various cell types, including a large number of tumor-associated macrophages (TAMs) [8,9,10]. The importance of these cells in gliomagenesis is increasingly appreciated. TAMs can make up half of the tumor mass [11]. These macrophages have become a focus of attention in recent years because they play a significant role in GBM progression and their increased presence is correlated with a poor prognosis [12]. TAMs can be derived from resident microglia, perivascular macrophages (PVMs), and infiltrating blood-derived myeloid cells [11,13,14]. They appear not only to promote tumor cell proliferation but also angiogenesis, and to suppress the normal functions of other immune cells [14], thereby contributing to glioma growth and resistance to treatment, which makes them an attractive target for new therapeutic approaches.

## 2. The Tumor Microenvironment in Glioblastoma

To better understand the roles of ionized calcium-binding adapter molecule 1 (Iba1)+ cells and CD163+ cells in their interplay with glioma progression, it is essential to appreciate the role of the TME. The TME of glioma comprises both non-immune and immune cells, including glioma stem cells (GSCs) [15], and is populated by a variety of immune cells, such as microglia and macrophages, T cells, B cells, natural killer cells, dendritic cells (DCs), and myeloid-derived suppressor cells (MDSCs) [16]. These immune cells form part of the immunosuppressive milieu of the TME, which is characterized by the production of immunosuppressive cytokines, activation of regulatory T cells, inhibition of CD4+ and CD8+ T cells, and reduction of major histocompatibility complex (MHC) II expression [11,17]. The non-immune cell component of the TMEs includes endothelial cells, neuroglia, neurons, and extracellular matrix in addition to tumor cells [16]. The blood brain barrier (BBB), which normally resides at the level of the tight junctions of central nervous system (CNS) endothelial cells, becomes leaky in malignant glioma and is transformed into the brain-tumor-barrier (BTB) [18,19]. Formation of the BTB, which is characterized by the loss of normal tight junctions and the development of structurally abnormal blood vessels, is stimulated by the upregulation of vascular endothelial growth factor (VEGF) and dependent angiogenesis. The altered permeability of the BBB is proportional to the degree of advancement of the malignancy [19,20]. Neurons, which are “overrun” by the glioma, can be found in diffusely infiltrated tissue. They are susceptible to the effects of cancer cells and actively contribute to cancer progression [21]. Glioma cells infiltrate the normal neuronal circuitry, secreting growth factors and neurotransmitters and even forming new ion channels and gap junctions [21]. Glioma cells also form synaptic connections with neurons, termed neuron-glioma synapses (NGS) [22]. These synapses are mediated by AMPA-type glutamate receptors, which foster glioma cell progression through the influx of calcium ions and downstream pathways.

The CNS is considered immunologically privileged and is home to a unique population of tissue-resident, specialized macrophages, the microglia. Bone marrow-derived macrophages (BMDMs) are absent from normal brain parenchyma but occur in small numbers around blood vessels that penetrate the tissue [23,24,25]. They physiologically undergo turnover with the periphery. Microglia originate from yolk sac-derived erythro-myeloid progenitors during brain development. They play a pivotal role in brain homeostasis and normal CNS functions. It has become clear in recent years that they have a distinct role in synaptic plasticity [26,27]. Thus, they are involved in learning and memory formation but also the phagocytosis of cellular waste byproducts and debris. Their phagocytic activity increases under certain disease conditions, and the morphology of activated microglia changes from highly ramified to cells that are more rounded and possess fewer processes [28,29,30,31].

BMDMs can infiltrate the CNS under pathological conditions, but only a relatively small number reside there under normal circumstances. These resident macrophages can be broadly categorized into three main groups: the already mentioned PVMs, meningeal macrophages, and choroid plexus macrophages. They are all located external to the blood brain barrier proper, which resides at the level of endothelial cell tight junctions. Their functions include preservation of the integrity of the CNS tissue barriers, surveillance of the tissue environment, e.g., when there is an infection, and assistance in recovery after injuries [32,33,34]. The activation of microglia and BMDMs is associated with the pathophysiologies of a variety of neurological conditions [14,35,36,37,38].

Taken together, the tissue microenvironment of glioblastoma is characterized by a complex ensemble of immune and non-immune cells, including microglia, macrophages, and neurons, which contribute to immunosuppression and cancer progression. Elucidating the roles of these cells and their interactions is essential for developing better therapies.

## 3. Overview of Iba1’s Structure, Function, and Its Expression in Normal and Pathologically Altered Brain Tissue

Iba1 was discovered by Imai et al. in 1996 [39] and found to be expressed by microglia. Iba1 is now commonly used as a marker for brain-resident microglia, although recent studies have shown that it can also be expressed by infiltrating macrophages [40,41,42,43]. This highlights the need for careful consideration of immune cell heterogeneity in the brain under pathological conditions. Iba1 is a 17kDa protein which belongs to the EF hand protein superfamily. It binds calcium ions and plays a crucial role in intracellular signaling involving Ras-Related C3 Botulinum Toxin Substrate (Rac) GTPase, phospholipase C-gamma (PLC-γ), and calcium signaling pathways [39,44]. The downstream effects of these pathways impact on cytoskeletal reorganization and phagocytosis [45,46]. In healthy brain tissue, microglia, in their normal state, monitor the local microenvironment, extending and retracting their processes while keeping their cell bodies stationary and away from other microglia [47].

Recent work shows that Iba1 is also involved in regulating synaptic development and function. Specifically, Iba1 plays a crucial role in the formation of excitatory synapses in the juvenile brain [48]. Studies using Iba1-deficient mice have demonstrated that Iba1 promotes synapse formation rather than limiting synaptic pruning. In Iba1-knockout mice [48], a reduced number of excitatory synapses results in changes in behavior, including impairments in object recognition memory and social interaction. Interestingly, Iba1 knockout models exhibit diminished microglial synaptic engulfment capacity, which may be a compensatory response to the early deficit in synapse formation. Synapse formation requires the extension or retraction of processes, which is accomplished by Iba1-mediated microglial protrusions that communicate with the surrounding environment. This process is thought to involve cytoskeletal reorganization within microglia, mainly via the dynamics of actin filaments [49]. The importance of Iba1 is illustrated in the formation and remodeling of parallel actin bundles, a crucial scaffold structure for motility protrusions such as lamellipodia and filopodia [49,50]. The mechanism behind microglial synapse engulfment, or synaptic pruning, involves Iba1 contributing to the formation of membrane ruffles, followed by the formation of a phagocytic cup through actin bundling [46]. This allows microglia to phagocytose and clean up unwanted synapses.

Iba1, also known as Allograft Inflammatory Factor 1 (AIF1), has emerged as a useful biomarker of immune activation [51]. Its role in cardiac allograft rejection was first highlighted by Ustans et al. [52], who observed that AIF1 was consistently expressed in chronically rejected cardiac allografts but absent in syngeneic grafts, underscoring its potential as a marker that promotes transplant rejection. This observation was later extended to kidney transplant models [53,54,55], where AIF1 was similarly associated with allograft rejection. However, one study [54] reported no such correlation, suggesting that its role may vary depending on the context or model used. Beyond transplantation, AIF1 has been implicated in a wide range of inflammatory and immune-related diseases, including rheumatoid arthritis, atherosclerosis, certain CNS disorders, and metabolic syndromes, as reviewed by Sikora et al. [51]. The expression of AIF1 is induced by inflammatory cytokines, including interferon-gamma (IFN-γ), and contributes to the modulation of immune responses, particularly by influencing the function of T helper 1 (Th1) cells, as evidenced by studies in colitis mouse models [56,57].

In the context of CNS disorders, AIF1 has also been recognized as a marker of microglial and monocyte activation, particularly in meningoencephalitis models [58,59], emphasizing its significance in immune modulation within the brain. Notably, in RAW 264.7, a macrophage cell line, when transfected with AIF1 cDNA, AIF1 is overexpressed, followed by a significant increase in the levels of IL-6, IL-10, and IL-12p40 upon bacterial lipopolysaccharide (LPS) stimulation [60]. These results suggest that AIF1 is not only a marker for macrophages and microglia but also supports the role of macrophages in immune responses.

Iba1 is a marker normally expressed in brain-resident microglia in both white and gray matter [61,62]. Notably, Iba1 can label both resting and activated microglia. However, our results align with other studies [63,64,65] that demonstrate a significant increase in Iba1 expression in microglia responding to glioma. This upregulation is accompanied by a deramification of microglia, leading to a more rounded, macrophage-like morphology, which typically exhibits a gradual transition forming a tissue gradient, as illustrated in Figure 1. Iba1 expression also increases in response to other tissue challenges, including reversible soft pathologies such as stress, and this increase is generally associated with increased microglial activation. Moreover, Iba1 plays a significant role in proinflammatory processes in the CNS [66,67,68]. Upon exposure to inflammatory stimuli, such as IFN-γ, Iba1 is upregulated, leading to microglial activation [56,57]. This upregulation promotes the secretion of proinflammatory cytokines and chemokines, including IL-6, thereby enhancing inflammatory responses [60]. At the molecular level, Iba1 interacts with the actin cytoskeleton and activates Rac GTPase signaling in microglia, which is crucial for cell motility and phagocytosis, highlighting its role in inflammation [44,45]. Furthermore, Iba1 expression is upregulated in response to ischemic injury, suggesting its participation in the sterile inflammatory response that follows ischemia, and potentially contributing to the ensuing pathophysiological consequences of ischemia [69].

Taken together, Iba1 (AIF1) plays an important role in regulating microglial function, synaptic development, and immune responses in the central nervous system.

## 4. Role of Iba1 in Glioblastoma Progression

The involvement of Iba1/AIF1 molecules in glioma progression, particularly in glioblastoma, is supported by accumulating evidence. For instance, Iba1+ microglia have been found to associate more closely with GSCs than CD163+ macrophages [63]. Furthermore, high levels of Iba1 expression are correlated with reduced patient survival, suggesting its potential as a prognostic marker [71]. Notably, in our view this association is not contradicted by the study of Woolf et al. [72], which had a limited sample size. Microglia marked by Iba1 contribute to the creation of an immunosuppressive environment that supports tumor growth. Microglia and other macrophages can exert pro-tumorigenic effects through the production of anti-inflammatory cytokines, such as transforming growth factor-β (TGF-β), which suppresses the normal function of Th1 cells [73]. A retrospective data analysis of 1270 glioma patients also demonstrated a potential correlation between Iba1 expression and GBM tumorigenesis [74]. TGF-β has been shown to promote GBM cell proliferation, invasion, angiogenesis, and immunosuppression [75]. The association between Iba1 and TGF-β is complex, with some evidence suggesting an inhibitory relationship [76]. Silencing Iba1 in an in vivo model demonstrated a significant increase in IL-10 levels in T cells under an inflammatory tissue condition [77]. Moreover, M2-polarized microglia have been shown to upregulate angiogenesis in the GBM microenvironment via supporting the transport of circKIF18A to human brain microvessel endothelial cells (hBMECs) [78]. In fact, microglia appear to be more important than peripheral macrophages in promoting angiogenesis [79]. Reducing the number of microglial cells had a significant negative impact on the formation of tumoral blood vessels [79]. Figure 2 shows Iba1 related pathways that are potentially involved in GBM progression.

The translocation of FOXC2 in human brain microvessel endothelial cells (hBMECs) has been shown to upregulate angiogenesis through both direct and indirect mechanisms. Furthermore, TGF-β has been found to be directly linked to FOXC2 via the EMT, suggesting an interplay between these factors in promoting angiogenic processes [91]. Having a strong correlation with TAM activation, GBM-associated polymorphonuclear leukocytes/granulocytes (GBM-hPMNL) in which the levels of pro-angiogenic cytokines and factors such as CXC motif chemokine ligand 2 (CXCL2), TEK, CD163, and hypoxia inducible factor-1α (HIF-1α) are increased have been identified [92]. These pro-angiogenic granulocytes were shown to facilitate the remodeling and regeneration of tumoral blood vessels in GBM [92]. A significant increase of CXCL2, a member of the CXC family, was demonstrated by Brandenburg et al. within GBM-bearing mouse brains, and its angiogenic effect was found to be even stronger than that of VEGF [79].

Glioma-derived macrophage colony-stimulating factor (M-CSF) plays a crucial role in modulating the behavior of microglia and macrophages within the tumor microenvironment. Not only does M-CSF regulate Iba1 translocation and microglial motility, it also acts as a potent inducer of angiogenesis through the upregulation of insulin-like growth factor-binding protein 1 (IGFBP1) [93]. The importance of IGFBP1 in this process is underscored by the finding that silencing IGFBP1 results in a significant decrease in tubular formation in endothelial cells [93]. Furthermore, IGFBP1 has been shown to promote GBM growth and survival by acting as an insulin-like growth factor, binding to prosurvival BAK and attenuating the growth-inhibitory effects of p53 [94]. Additionally, IGFBP1 is a potent activator of nitric oxide production through the phosphoinositide 3-kinase (PI3K) signaling pathway, which potentiates angiogenesis [94,95]. The mechanism by which M-CSF exerts its effects on microglia and macrophages involves the translocation of Iba1 to membrane ruffles [46], where it co-localizes with actin filaments and promotes motility and migration [45]. In the absence of M-CSF, Iba1 remains in the cytoplasm, and its function is impaired. The presence of M-CSF, on the other hand, triggers the PI3K-NFκB pathway [93], leading to an increase in M-CSF secretion by GBM cells. This increase in M-CSF levels has a dual effect on the tumor microenvironment. Firstly, it upregulates angiogenesis through an increase in VEGF [96] and IGFBP1 [93]. Secondly, M-CSF facilitates the co-localization of Iba1 with actin filaments, promoting the migration of microglia and macrophages within the tumor microenvironment and aiding tumor progression [46,97].

Our recent work suggests that there is a potential crosstalk between GSCs and Iba1+ TAMs [63]. IL-33 has been shown to act as a bidirectional messenger between these cells: GSCs release IL-33, which binds to ST2 receptors on TAMs [98], increasing downstream STAT3 protein phosphorylation and upregulating IL-6 and LIF [99]. This actively recruits TAMs, including Iba1+ TAMs [100], to the TME, and switches TAMs from an anti-tumorigenic state to a pro-tumorigenic, anti-inflammatory phenotype that promotes gliomagenesis. Interestingly, Iba1+ TAMs can also produce IL-33, which in turn is expected to feed back to GSCs, allowing them to maintain self-renewal and sustain stemness, supporting GBM progression [63,100].

Taken together, the involvement of Iba1/Aif1 molecules in glioma progression highlights their role in creating an environment that supports tumor growth and suggests their potential as a therapeutic target.

## 5. Iba1 as a Potential Future Therapeutic Target in Glioblastoma

Therapies that specifically target tumor-associated Iba1+ve microglia may represent a promising new strategy for treating GBM. Silencing Iba1 in the BV2 cell line using Iba1-small interfering RNA (Iba1-siRNA) has shown significant efficacy in reducing the migratory capacity, proliferation, and cell adhesion of BV2 microglia [101]. Interestingly, the ability of BV2 microglia to phagocytose was upregulated by silencing Iba1. This is due to the increase in the expression of P2X7 [101], a scavenger protein responsible for phagocytosis [102]. Silencing Iba1 could potentially also be exploited to reduce glioma cell migration [103].

Interestingly, Iba1-siRNA [101] has been shown to upregulate P2X7, significantly increasing the radiosensitivity of experimental GBM and resulting in significant tumor volume reduction after radiotherapy [104]. However, the method for delivering siRNA into CNS parenchyma poses a substantial challenge, and the efficacy is low even when siRNA is delivered directly into the cerebrospinal fluid (CSF) [105]. This technical difficulty could potentially be overcome by conjugating the desired siRNA to a lipophilic or hydrophobic agent, which facilitates dispersion of siRNA to the entire CNS parenchyma [106]. However, siRNA used with conjugates could be associated with neurotoxicity [107]. Thus, while Iba1-siRNA may potentially slow GBM progression, more studies are needed to make this new technology useful in practice.

The significance of Iba1 in GBM research is outlined in Table 1, which also identifies key areas where further investigation is needed to address existing knowledge gaps.

## 6. CD163, a Macrophage Scavenger Protein Showing Differential Expression in Glioblastoma

CD163, a scavenger receptor and member of the scavenger receptor cysteine-rich (SRCR) superfamily, is exclusively expressed in macrophage/monocyte lineages [124]. The extracellular component of CD163 consists of nine SRCR domains that are available for the binding of ligands, including Hb-Hp complexes [124,125,126]. Specifically, SRCR domains 2 and 3 mediate the binding affinity of CD163 to the Hb-Hp complex, which regulates crucial cellular self-defenses such as the prevention of oxidative damage and anti-inflammation [127,128,129]. Hp-bound Hb is recognized by CD163+ macrophages, which triggers Hb-Hp complex internalization and the subsequent degradation and recycling of iron [125]. The anti-inflammatory effects of CD163 are also triggered by the endocytosis of the Hb-Hp complex, which is preceded by the release of the anti-inflammatory cytokine, IL-10 [130]. In addition to IL-10, IL-6 has also been shown to induce the expression of CD163, while Interleukin-4 (IL-4), tumor necrosis factor-α (TNF-α), IFN-γ, and bacterial LPS attenuate CD163 expression [131].

PVMs are a special population of macrophages that reside within the perivascular spaces, where they are separated from both the actual BBB and the brain parenchyma by basement membranes [132,133]. These cells are crucial in maintaining the homeostasis of the CNS environment, aiding in immune surveillance, antigen presentation, and maintaining the BBB’s integrity [132,134]. Interestingly, CD163 is expressed in the PVMs of the CNS in normal physiological conditions but not in microglia [135,136]. It was demonstrated in a simian immunodeficiency virus enchephalitis (SIVE) animal model that normal and acutely infected animals express CD163 in PVMs only [137]. However, in the same study, chronically infected animals showed CD163 expression in activated microglia as well [137]. These findings are supported by Pey et al.’s work [138]. CD163 expression has been described in several diseases [138,139], Table 2.

## 7. Regulation of CD163 Expression and Its Implications for the Immune Response

CD163 expression is regulated in a complex fashion in response to proinflammatory and anti-inflammatory stimuli. Bacterial LPS is a Toll-like receptor 4 (TLR4) agonist that exerts important immune mediating functions via the TLR4-nuclear factor -κB (NFκB) pathway by releasing proinflammatory cytokines such as IL-1β, IL-6, IL-8, and TNF-α [144]. The release of these cytokines downregulates the expression of CD163 both directly [145] and via activation of metalloprotease enzyme 17 (ADAM17) [144]. An in vitro study on human adipose tissue suggests that LPS does not alter the level of CD163. Instead, LPS significantly increases sCD163 (soluble CD163] [146]. The activation of the downstream NF-κB pathway can also be achieved by the upregulation of a tumor necrosis factor (TNF)-like weak inducer of apoptosis (TWEAK) [147,148]. TWEAK seems to be a promising target to prevent CD163+ macrophages from expressing proinflammatory cytokines.

CD163 is regarded as a marker of M2 macrophages [149]. CD163+ TAMs secrete pleiotrophin (PTN), which binds to its receptor on GSCs, protein tyrosine phosphatase receptor type Z1 (PTPRZ1), promoting self-renewal and sustainability that support tumor progression [150]. Like Iba1, CD163 expression can also be enhanced by increased IL-33 levels and its downstream IL-6/STAT3 signaling pathway, contributing to TAM recruitment and an anti-inflammatory shift [100]. The secretion of IL-6, facilitated by hypoxic glioma-derived exosomes, has an additive effect on increasing CD163 and IL-10 expression via the same STAT3 pathway, thereby enhancing tumor progression [151]. CD163+ TAMs have also been implicated in the release of C-C motif chemokine ligand 2 (CCL2), another essential cytokine for the recruitment of MDSCs and regulatory T cells [152].

## 8. Role of CD163 in Antigen Presentation and Cytokine Production

CD163 also functions as a receptor for TWEAK [153]. Although not fully understood at present, the internalization of TWEAK by CD163 can have both beneficial and detrimental effects in the event of cellular stress or injury. An in vivo study suggested that an increase of TWEAK expression can be induced by CD163 gene deletion [154]. Another in vivo study pointed out that CD163 deficiency is associated with atherosclerotic plaque formation [155]. The effects of TWEAK upregulation in the context of metabolic disease have also been discussed, emphasizing the importance of CD163 as a chief regulator of the immune response and its downstream signaling pathways [156]. CD163+ macrophages, especially PVMs, are believed to have important immunomodulatory functions. A recent single-cell RNA sequencing study revealed that anti-inflammatory gene expression was upregulated as GBM progresses [157]. Specifically, the transcripts of CCL2, a key cytokine responsible for macrophage recruitment [158], are decreased in GBM, resulting in the decrease of C-C motif chemokine receptor 2 (CCR2) activation and reducing the infiltration of monocytes, transitioning from proinflammatory to immunosuppressive signaling. Pericytes are the mesenchymal cells that specialize in angiogenesis and vessel stabilization in CNS [159]. PVMs have been shown to promote the expansion of a pro-angiogenic niche, primarily composed of pericyte-like mesenchymal cells, through the expression of platelet-derived growth factor-C (PDGF-C) [160]. This expansion of pro-angiogenic factors provides grounds for GBM progression. MHC I is ubiquitously expressed in all nucleated cells, which is important for promoting a CD8+ T-cell response when brain tissue is threatened by infectious agents [161]. One CX3CR1 mouse model, where the H2-Db component of MHC I receptors was deleted in PVMs, showed a decline in the number of infiltrating CD8+ T cells in mouse brains under infectious conditions [162]. Although CD163 has not been shown to be directly involved in antigen presentation to CD4+ T helper cells, CD163 and MHC II, as well as co-stimulatory molecules (e.g., CD80, CD86 and CD40) have been found to be co-expressed in PVMs [163]. ADAM17, a member of a metalloprotease family also known as tumor necrosis factor α-converting enzyme (TACE), is responsible for cleaving various surface proteins, including CD163 [164]. ADAM17 cleaves the ectodomain fragment of CD163, which releases sCD163 that can enter the bloodstream [165]. An initial increase in sCD163 that is directly influenced by the level of inflammatory stimuli has been described [165]. Heightened levels of ADAM17 and sCD163 have been observed in stroke patients [166]. Lymphocyte proliferation can be halted by sCD163 [166]. Another study [164] demonstrated downregulation of CD163 expression induced by an upregulation of ADAM17 during a viral infection.

## 9. Expression of CD163 by Macrophages and Microglia in Glioma

CD163 is predominantly found in BMDMs and is generally absent from normal resident microglia [43,137,167]. Chen et al. have reported that an increased expression of CD163 is inversely correlated with survival in gliomas, and inhibition of CD163 led to a significant reduction in GSC stemness [168]. CD163 has been suggested to be a promising biomarker and therapeutic target for GBM [169].

CD163 is more highly expressed in glioblastoma tissue than in normal brains or low-grade gliomas (LGGs) [43]. This expression is directly related to the grade of malignancy as well as poorer survival for both male and female glioblastoma patients [140,169]. Single-cell transcriptomic studies have identified distinct markers for microglia (P2RY12 and TMEM119) and TAMs (CD14 and CD163), although there is some overlap in marker expression [72]. In glioblastoma, Iba1-positive cells (including both microglia and macrophages) are often found demarcating necrotic areas. In contrast, CD163 can show extensive extracellular deposition within necrotic tumor areas [170]. The differential localization of these markers in relation to necrosis suggests different roles for myeloid TAMs and microglia [70]. A recent single-cell RNA sequencing (scRNA-Seq) analysis study revealed a new CD163+HMOX1+ cell population, which is responsible for the inhibition of T-cell responses via IL-10 secretion [121]. Treatment with an anti-programmed death-ligand 1 (PD-L1) antibody has demonstrated a significant reduction of CD163+ macrophage infiltration in glioma, implying a potential relationship between CD163 expression and immune checkpoint mechanisms [171]. In addition, CD163+ cells may play a role in regulating angiogenesis and inflammation within the tumor microenvironment. Overall, the expression of CD163 on microglia/macrophages appears more common in high-grade glioma tissue. Observations in other diseases (Table 2) and especially experimental work on the CD163 rat homologue, ED2, support the view that microglia are principally capable of expressing the CD163 molecule, but microglia in healthy brain tissue and most disease conditions do not [172].

The following considerations assist in evaluating the complex tissue expression of Iba1 and CD163 in glioblastoma (Figure 3 and Figure 4). This evaluation can be supported by AI-based identification of macrophages and microglial cells, which we are currently developing (Figure 5 and Figure 6). Resident microglia typically exhibit a ramified morphology characterized by branching cell processes that not infrequently extend perpendicularly. If CD163-positive cells in the brain display this morphology, even if only partially preserved with shorter, stouter processes, it suggests that these cells are resident microglia that have acquired CD163 expression. Similarly, one would expect microglia-derived macrophages to retain at least some of their processes within the tissue, as visible in Figure 4A,B. Additionally, microglia do not form gap junctions with each other, unlike neuroglial cells, but maintain a distance from one another. If CD163-positive cells are spaced throughout the tissue, rather than being predominantly localized around blood vessels, it indicates that they are likely not recently infiltrated peripheral macrophages. In brain tissue heavily infiltrated by glioma, these criteria become less reliable. However, blood-derived macrophages typically lack the cell process stubs characteristic of microglia-derived macrophages (Figure 4C–E) and are more often found in the vicinity of blood vessels.

## 10. Studying the Glioblastoma TME at a New Level: Using Iba1 and CD163 for Artificial Intelligence-Assisted Detection of Microglia/Macrophages

Recent evidence indicates that microglia and blood-borne macrophages exhibit distinct associations with different components of the TME. Specifically, Iba1-immunoreactive microglia are more commonly found in viable tumor areas, whereas CD163-expressing macrophages, likely of recent bone marrow origin, predominate in necrobiotic and necrotic tissue areas of glioblastoma [70].

Microglia have been called a sensor of pathology of the CNS and there is a strong correlation between their morphological phenotype and state of activation. Thus, being able to analyze entire tissue samples in detail by identifying all morphological microglia and macrophage subtypes may allow us to uncover hidden patterns in these populations. We have started to establish the methodology to tackle this problem and would like to illustrate the basic methodology, as it is very pertinent to the topic of this in-depth review. The approach is currently confined to macrophage-like (fewer process bearing) iba1+ microglia and CD163+ cells, but the plan is to extend the method to include a large variety of microglial morphological phenotypes.

PathoFusion, our recently developed artificial intelligence (AI) framework, enables us to undertake this work (Figure 5 and Figure 6). PathoFusion is an open-source AI platform designed to annotate, train, and automatically identify key morphological features in whole-slide images of tissue samples [173]. At the core of PathoFusion lies a bifocal convolutional neural network (BCNN) that combines the parallel use of small and large image tiles. This combination allows PathoFusion to classify morphological and immunohistochemical characteristics across stained sections with high accuracy, providing a much deeper analysis of tissue samples than it is possible to obtain through conventional microscopy work. PathoFusion automatically detects and profiles individual cells, i.e., it is capable of identifying specific cell types, helping the user analyze their spatial relationships and suggest potential interactions. Furthermore, immunohistochemical markers can be linked to histological, cellular, and even subcellular anomalies. Uncovering the relationship between molecular markers and tissue structures can provide insights into disease mechanisms. In addition, comprehensive cross-modality analyses can be performed integrating data from multiple sources, potentially including electron microscopy and specialized diagnostic imaging methods to gain a deeper understanding of tissue biology and pathology. PathoFusion’s BCNN is designed to learn quickly and efficiently, requiring minimal training samples to produce high-quality results [173,174]. This means that morphology experts, such as pathology consultants, can define features of interest and transfer their knowledge to the AI system, reducing the time demands for annotation and enabling routine AI-driven feature recognition tasks. By making use of the power of machine learning and AI, PathoFusion has already become a very useful tool, and the platform offers new opportunities for discovery and exploration in both basic and clinical research.

## 11. TAMs Undergo Metabolic Reprogramming

Metabolic reprogramming and the response to hypoxia are considered main mechanisms for cancer invasion and immunosuppression. Rapid tumor progression requires high amounts of energy and nutritional supply. To meet this energy demand, glioma cells undergo “metabolic rewiring”, in which the metabolism of nutrients is altered. For example, glioma cells shift toward aerobic glycolysis (Warburg effect), in which glucose is converted to lactate even in the presence of oxygen [175]. As a result, energy metabolites accumulate and provide the nutritional basis for rapid cell division and biomass production. Lactate is the main product of the altered glycolysis, which has also been shown to be utilized by TAMs [175]. TAMs also undergo metabolic reprogramming in response to hypoxia and cytokines (e.g., IL-10) in the TME [110]. Hypoxia triggers the transcription of the HIF-1alpha gene, which forces a switch of energy production to glycolysis. If enough oxygen is available, HIF-1alpha is hydroxylated in the presence of oxygen and prolyl-4-hydroxylase 2 (PHD2), which is subsequently degraded in proteasomes and recycled [176,177]. However, if oxygen saturation is low, PHD2 is inhibited, causing the accumulation of HIF-1alpha. When HIF-1alpha is elevated, binding to HIF-1beta occurs, stimulating the transcription of pro-angiogenic genes such as VEGF and epidermal growth factor receptor (EGFR). This leads to the formation of new blood vessels [178]. Under normal conditions, HIF-1alpha upregulation protects against hypoxia. However, HIF-1alpha is also promoted in the TME, where it supports cancer cell survival, invasion, and migration [179]. The accumulation of lactic acid and hydrogen ions lowers the cellular pH, which can also promote tumor invasion and local tissue destruction by the upregulation of metalloproteinases [180]. This is often accompanied by glutamate excitotoxicity-induced neuronal death due to the decrease in pH [181]. The combined effect of lactate production and a low pH level further cause immunosuppression in the TME resulting from the dysfunction of T cells and natural killer cells, which supports the survival of tumor cells and skews TAMs toward a tumor-supportive immunosuppressive state [110,182].

## 12. TAMs Weaken the BBB, Supporting GBM Invasion and Growth

The BBB serves as a crucial boundary, safeguarding the brain against extraparenchymal insults while maintaining the homeostasis of its microenvironment. The extracellular matrix (ECM) is a vital scaffold for BBB cells. It is susceptible to damage, which can compromise vital interactions between endothelial cells and pericytes [183]. Such damage is associated with tissue factors that are released from TAMs such as C-C motif chemokine ligand 5 (CCL5) and matrix metalloproteinases (MMPs). CCL5 released by microglia upregulates MMP2 in GBM via the calcium dependent pathway, which degrades the ECM of the BBB and promotes the invasion of GBM into the brain parenchyma [184]. The expression of MMP2 can also be induced by microglia directly via the activation of TGF-β and downregulation of tissue inhibitors of metalloproteinases (TIMP) [11,185]. Other MMPs such as MMP9 and MMP14 have also been implicated in the breakdown of the ECM [186]. VEGF, a pro-angiogenic factor released by TAMs upon GBM invasion, stimulates the formation of new blood vessels following binding to VEGF receptors [187]. This is achieved by the production of MMPs that break down the ECM, which facilitates cell migration to the site and promotes endothelial proliferation. Therefore, the release of VEGF makes the BBB more porous, which enables tumor cells to invade and escape. However, Sarkaria et al. disagreed with the popular belief that GBM brains have a uniformly disrupted BBB. Instead, they found that all GBM patients have tumor regions with an intact BBB [188]. This finding underscores the fact that many anticancer medications, which have a limited ability to cross the BBB, remain ineffective against GBM [188].

## 13. Role of Microglia and Macrophages in Glioblastoma Progression

Microglia/macrophages, which express Iba1 and CD163 markers, likely play a significant role in shaping the tumor microenvironment and contributing to glioma development. In GBMs, microglia/macrophages are present in large numbers and can account for up to around half of the tumor mass [12,189]. These cells are not just passive bystanders. They actively contribute to immunosuppression, tumor growth, and progression. TAMs are capable of influencing glioma growth by altering the recruitment of immune cells, downregulating microglia/macrophage defense mechanisms, and secreting factors that support tumor progression [9]. The upregulation of secreted phosphoprotein 1 (Spp1), a glycophosphoprotein expressed in immune cells like BMDMs and T cells that is responsible for chemotaxis and migration, has been reported in glioblastoma cells [190]. P-Selectin (SELP) is an adhesion molecule that is responsible for immune cell recruitment [191]. Evidence exists that SELP, which is produced by glioma cells, together with its ligand P-Selectin Glycoprotein Ligand-1 (PSGL-1), is able to reduce the phagocytic activity of TAMs and increase the release of the immunosuppressive cytokines TGF-β and IL-10 [9]. The secretion of IL-10 can contribute to the metabolic reprogramming of TAMs, stimulating downstream HIF-1α angiogenesis and tumor progression [110]. The elevated expression of Na+/H+ exchanger isoform 1 (NHE1), which is involved with microglial activation, is positively correlated with glioma proliferation and migration [80]. On the other hand, the inhibition of NHE1 has been shown to improve anti-tumor immunity in TME by rejuvenating oxidative phosphorylation and downregulating pro-tumorigenic factors released by microglial cells through decreased intracellular pH [192,193].

## 14. Potential Future Directions for Unravelling the Functions of TAMs in GBM

Recent studies have provided promising evidence for the usefulness of liquid biopsy in retrieving circulating tumor DNA (ctDNA), a type of DNA that is tumor-specific [194]. The detection of ctDNA has been shown to greatly assist in the diagnosis of solid tumors, such as colorectal, breast, and lung cancers [195]. However, the BBB poses a significant challenge in the case of GBM, with only approximately 1% of GBM ctDNA detectable in plasma [196]. An alternative approach to overcome this limitation involves collecting samples from CSF, which may offer improved detection rates [195,197]. Despite its potential as a diagnostic tool, liquid biopsy has limitations in providing insights into the dynamic nature of the TME in GBM, particularly with regards to real-time visualization of immune cell infiltration [198].

Molecular barcoding (MB) is an emerging next-generation sequencing technology that enhances accuracy and sensitivity in detecting rare DNA variants, offering significant improvements over traditional methods [199]. Through the use of amplification techniques, MB enables the high-throughput identification of mutations from ctDNA, even when present at very low concentrations in plasma [199]. However, despite these advancements, MB analysis of ctDNA has inherent limitations, notably its inability to comprehensively reveal the complex immune compositions of the TME. Fortunately, liquid biopsy approaches utilizing exosomal circular RNAs (circRNAs) have shown promise in addressing this challenge. circRNAs are particularly valuable as they reflect, in real time, how glioma cells and immune cells respond to environmental changes, thereby providing a dynamic snapshot of the TME [200].

Furthermore, the methylation signature has emerged as an innovative tool for assisting in the early detection of malignancy [201]. The term “methylation signature” refers to the specific pattern of DNA methylation, characterized by the addition of a methyl group to cytosine residues, which yields a unique epigenetic fingerprint for a particular disease or cell type [202]. Recent research has identified distinct methylation signatures associated with long-term survivors of GBM, defined as patients who exhibit extended survival beyond typical expectations after diagnosis [203]. These findings suggest that methylation signature analysis may offer valuable insights into the biological characteristics distinguishing this subgroup of patients. Moreover, this technology holds promise for investigating the origins of TAMs, characterizing their diverse states, and detecting epigenetic changes in TAMs induced by glioma cells [204].

Patient-derived xenografts offer unique insights into cancer pathobiology and therapeutic target discovery. Recently, a mouse model [205] was used to study how different types of anesthetics (volatile vs intravenous) affect breast cancer metastasis. This model successfully replicated the effects of anesthetic agents, shedding light on their underlying mechanisms. Similarly, other xenograft models, such as those established with colorectal cancer [206] and non-small cell lung cancer cells [207], have yielded valuable information on human cancer biology in vivo. In the context of GBM, several xenograft models have been developed, including heterotopic (implanted outside of the CNS) and orthotopic (implanted within the CNS) implantation approaches, as reviewed by Gómez-Oliva and colleagues [208]. While heterotopic models facilitate direct observation of GBM development and provide a platform for preclinical drug testing, they fail to accurately recapitulate the microenvironment of the CNS [208]. Consequently, orthotopic xenografts have gained popularity, as they more closely mimic actual GBM conditions and interactions with immune cells within the CNS, despite presenting technical challenges [209].

## 15. Novel Therapeutic Agents Targeting TAMs

Therapeutic agents aimed at targeting TAMs in the TME to combat GBM progression have recently gained significant attention. M-CSF, a cytokine essential for TAM differentiation and GBM progression [210], has emerged as a promising target candidate for reducing GBM growth. Inhibiting M-CSF or its receptor, colony-stimulating factor-1 receptor (CSF-1R), has shown promise in preclinical GBM models [211]. A small oral agent, PLX3397, was developed to directly target CSF-1R, demonstrating efficacy in GBM preclinical studies [212]. However, human clinical trials (NCT01349036, NCT01790503) have reported limited efficacy [213,214]. Another CSF-1R inhibitor, BLZ945, has been shown to improve survival rates [211] and enhance radiotherapy efficacy in glioma mouse models [215]. In humans, BLZ945 can augment the efficacy of anti-PD-1 therapy against GBM by attenuating the immunosuppressive polarization of CD163+ macrophages [216]. However, prolonged treatment with BLZ945 has yielded no additional benefits [217].

CD47, a “don’t eat me” signal on tumor cells [218], has also been investigated as a potential target. An in vivo study found increased GBM phagocytosis following anti-CD47 treatment [219], whereas another model demonstrated minimal effects on glioma growth with anti-CD47 monotherapy [220]. Moreover, anti-CD47 treatment can induce hematological toxicity [221]. To date, human studies on anti-CD47 therapy are limited, and further research is needed to fully understand its potential. The TGF-βR1 inhibitor SB431542 has been shown to impair tumor growth by 75% in a mouse model [222], but the translatability of these findings to human studies remains uncertain [223,224].

## 16. Conclusions

The role of microglia and macrophages in the tumor microenvironment of glioblastoma is complex, contributing to immunosuppression, tumor growth, and progression. These TAMs modulate immune cell recruitment, suppress defense mechanisms, and secrete factors promoting tumor expansion. The targeting of microglia and macrophages is a promising novel therapeutic strategy, with potential approaches including inhibition of NHE1, modulating the M1/M2 balance, and developing BBB-penetrating therapeutics. To overcome the challenges of glioblastoma treatment, AI-assisted analyses can provide valuable insights into the complex interactions between microglia, macrophages, and the tumor microenvironment. By applying methods such as single-cell analysis, spatial analysis, network analysis, and predictive modeling, additional targets for GBM treatment should become available, enabling the development of more effective therapies. These will likely be combination therapies that focus on multiple aspects of the tumor microenvironment. A comprehensive understanding of microglia and macrophage behavior in GBM appears essential for improving treatment outcomes.

## Figures and Tables

**Figure 1 cancers-17-01457-f001:**
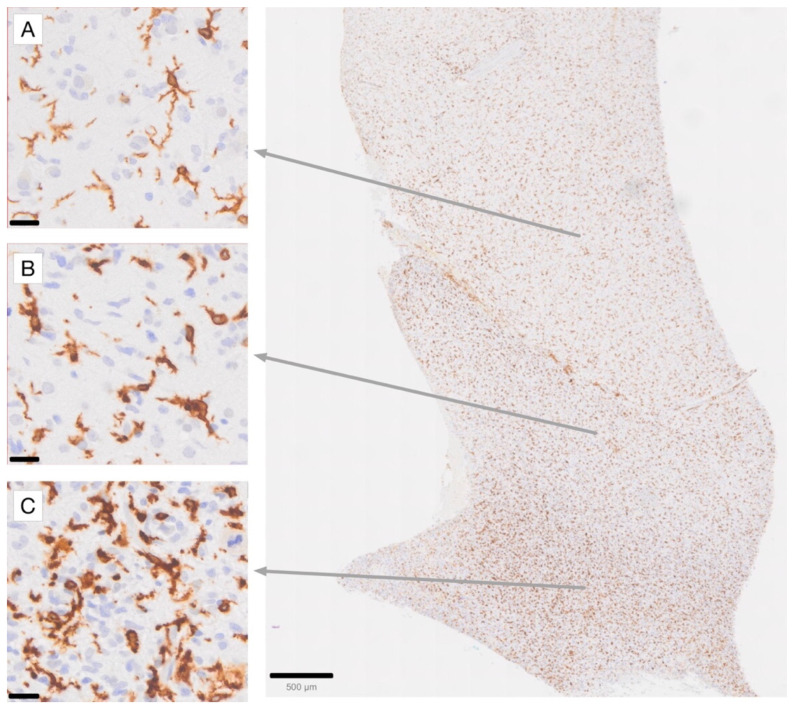
Immunohistochemical staining for Iba1 in a high-grade glioma (glioblastoma) case, demonstrating the gradient of microglial activation in areas of diffuse tumor infiltration [63,70]. (**A**) Highly ramified microglia with moderate Iba1 labeling intensity. (**B**) Microglia with stouter cell processes and increased Iba1 labeling intensity. (**C**) Densely tumor-infiltrated brain area with an increased number of microglia exhibiting intense Iba1 labeling. Note that while some macrophages, such as those in perivascular locations, may also express Iba1 in a normal state, the gradient of activation observed among parenchymal cells, which are more evenly spaced, allows for the identification of Iba1-immunoreactive macrophages in this location as being derived from activated microglia. Scale bar in (**A**–**C**): 20 µm. Scale bar in the overview photograph on the right: 500 µm.

**Figure 2 cancers-17-01457-f002:**
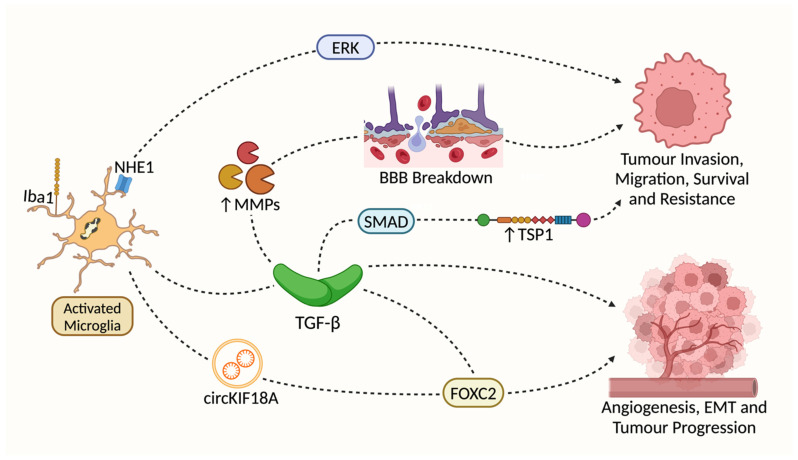
Pathways proposed to be involved in Iba1’s effects on GBM progression. Both Iba1 and Na+/H+ exchanger isoform 1 (NHE1) are associated with microglial activation in the context of glioma progression [80,81]. However, there is no direct evidence linking Iba1 to NHE1-initated pathways. Transforming Growth Factor-β (TGF-β) is significantly upregulated in Iba1+ microglia [80,82,83]. TGF-β stimulates the expression of Matrix Metalloproteinases (MMP2 and MMP9) expression [11,84,85], which may support GBM invasion via the breakdown of the BBB. TGF-β may also activate downstream SMAD-dependent signaling [83,86,87], promoting GBM invasion and treatment resistance via an increase in Thrombospondin 1 (TSP1) expression [88,89]; M2-polarized microglia may activate the translocation of Forkhead Box Protein 2 (FOXC2) to Human Brain Microvessel Endothelial cells (hBMECs) both directly (via the translocation of circKIF18A) [78,90] and indirectly (by TGF-β) [91], which may promote angiogenesis, the epithelial-mesenchymal transition (EMT) and overall tumor progression.

**Figure 3 cancers-17-01457-f003:**
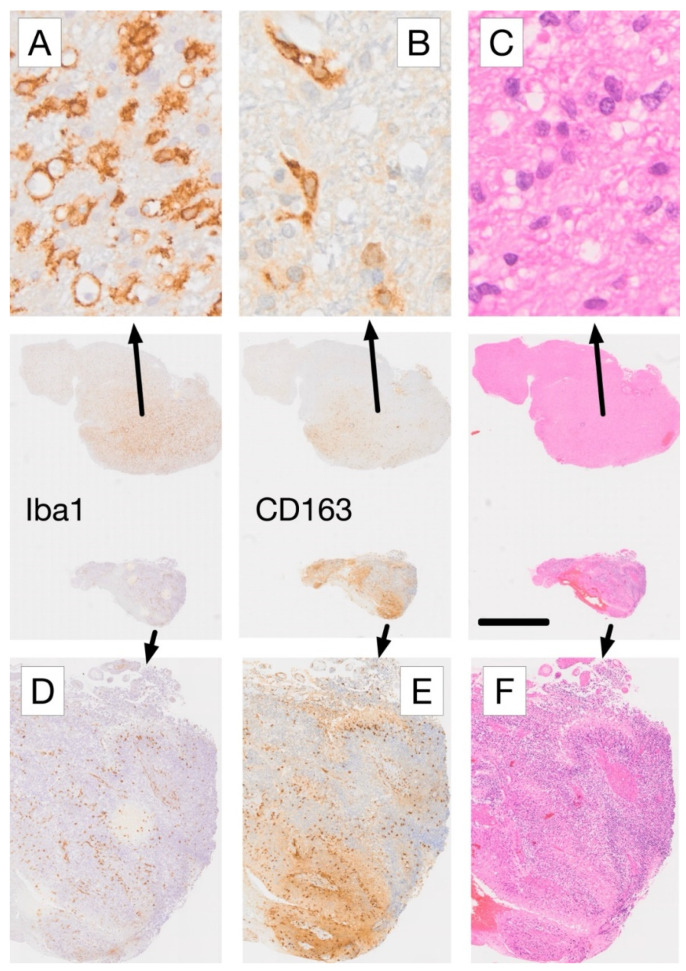
Comparative immunohistochemical analysis of Iba1 and CD163 expression in adjacent tissue sections from a glioblastoma biopsy. (**A**) In areas with low-density glioma infiltration, microglia-derived cells with rounded morphology and strong Iba1 expression (arrow in A) lack corresponding CD163 staining, whereas mainly perivascular cells (macrophages) exhibit CD163 labeling (**B**). (**C**) Haematoxylin and eosin (H&E) staining confirms low density tumor cell infiltration in this region. In contrast, areas with high-density glioma infiltration (**D**–**F**) show a reversed expression pattern, with more CD163+ cells (**E**) than Iba1+ cells (**D**). The presence of palisading necrosis in this tissue region (**F**), characterized by myeloid macrophages dominance [70], is consistent with the observed difference in marker expression. Scale bars: 20 µm (**A**–**C**), 2 mm (center row overview images), and 250 µm (**D**–**F**). For more information on the IHC labeling used please refer to [70].

**Figure 4 cancers-17-01457-f004:**
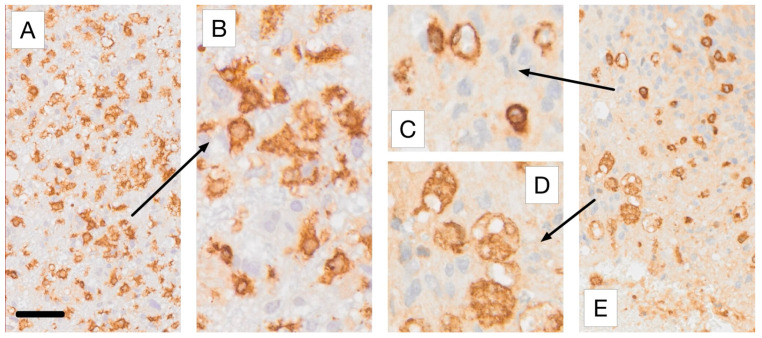
The main macrophage phenotypes encountered in Iba1 and CD163 immunostaining of glioblastoma tissue. All samples represent higher magnifications of the regions of interest selected for Figure 3A,E, respectively. (**A**,**B**) show Iba1 immunoreactive microglia-derived macrophages with retained short processes, shown at lower (**A**) and higher (**B**) magnification. (**C**,**D**) Higher-magnification views of the regions marked by the arrows in Figure 3E, illustrating CD163+ myeloid macrophages that typically possess far fewer cell processes than microglia-derived phagocytes. (**D**) shows foam cells. Scale bars: 50 µm (**A**,**E**) and 20 µm (**B–D**). For more information on the labeling used please refer to [70].

**Figure 5 cancers-17-01457-f005:**
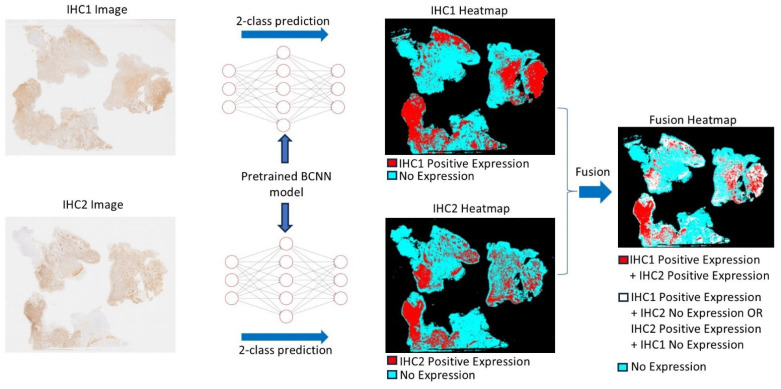
The diagram illustrates how the Bifocal Convolutional Neural Network (BCNN) is utilized to recognize specific cell types, with a focus on round microglia and macrophages in our example. Two whole-slide images (WSIs) of adjacent tissue sections immunohistochemically stained using two different antibodies are processed using the BCNN model, where two distinct heatmaps are produced via a 2-class prediction method. The heatmaps of the two immunohistochemistry (IHC) images are then fused, creating a fusion heatmap. The fusion heatmap indicates the correlation between the two markers (red), areas of non-correlation (white), and the area of no expression of either marker (turquoise). It is essential to note that all representations refer only to the selectively detected cell populations of interest and not to all immunohistochemical staining results. This critical focus is made possible by AI assistance and would not be achievable using conventional microscopic methods (for further explanations see [63]).

**Figure 6 cancers-17-01457-f006:**
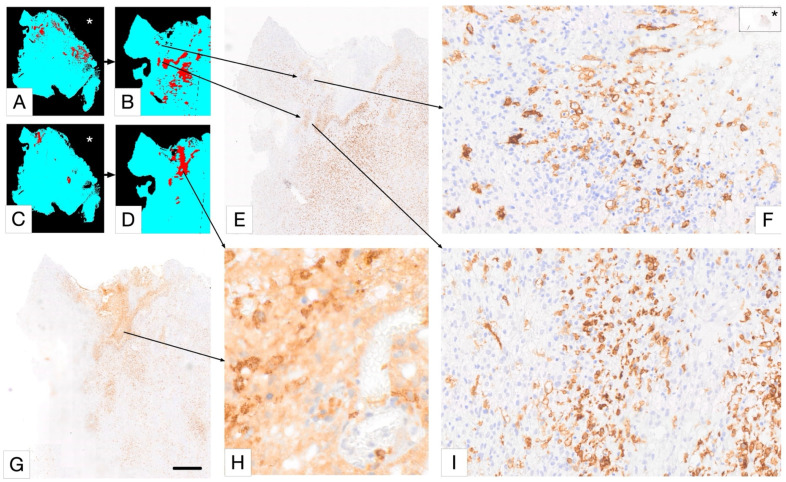
Selective detection of macrophage/microglia phenotype using PathoFusion. In this Figure, (**A**,**B**,**E**,**F**,**I**) represent Iba1 immunostaining, and (**C**,**D**,**G**,**H**) represent CD163 immunostaining. Only “deramified” microglia are shown in (**A**,**B**); the rest of the microglia are ignored for this analysis. Scale bars: 800 µm (**E**,**G**), 50 µm (**F**,**I**), and 20 µm (**H**). The asterisks in (**A**,**C**,**F**) indicate WSI views of the same case.

**Table 1 cancers-17-01457-t001:** Iba1 in glioblastoma.

Topics	Key Findings	Knowledge Gaps
Iba1 as a Biomarker	High Iba1 levels in tumor samples has been suggested to correlate with shorter survival in GBM patients [71,74].	More studies are required to validate Iba1 as a prognostic marker.
Molecular Mechanisms	Secretion of IL-10 contributes to metabolic reprogramming of TAMs [108,109], upregulating HIF-1α dependent angiogenesis and tumor progression [110,111,112].	How intercellular interactions within the TME influence metabolic pathways, treatment resistance, and tumor heterogeneity, specifically: cell-cell communication, paracrine signaling, glycolysis and glucose metabolism, lipid and amino acid metabolism, mitochondrial function, metabolic reprogramming, immune suppression, clonal evolution, and epigenetic reprogramming.
Tumor Microenvironment	Iba1 is associated with the TGF-β signaling pathway [82], contributing to immunosuppression in TME; TGF-β activation facilitates the Ras/Raf/MEK/ERK signaling pathway [113] and PI3K/AKT/mTOR pathway [114,115].	The mechanisms by which GBM induces immunosuppression in Iba1+ cells are not well understood; the exact mechanisms and underlying effects of the Iba1-associated signaling pathway are unclear.
	GBM microenvironment promotes immunosuppressive activation states of Iba1+ cells [9,73,75,77,84,86,88,110,116]	There is limited understanding of the response to hypoxia of both microglia and BMDMs.
Angiogenesis	Iba1 upregulates angiogenesis in GBM microenvironment via translocation of FOXC2 [78,90] and could be linked to TGF-β [91].	Differentiation between pro-angiogenic cytokines secreted by microglia and BMDMs.
	TAMs accumulate in areas of tumor lacking oxygen, whereas M1-like TAMs localize in normoxic tissue areas [117].	Interaction with other cell types promoting new vessel formation.
Microglia Phenotype and Heterogeneity	Microglial shift from “M1” to “M2” phenotype in GBM microenvironment [9,116,118,119,120], though dichotomic categorization oversimplifies the heterogeneity of microglia / macrophage phenotypes [27,121,122].	Distinct transitory states of microglia are yet to be clarified.
	Iba1+ TAMs exhibit intra-tumoral heterogeneity [41].	A more comprehensive understanding of microglial states through integration of transcriptomic data with epigenomic, proteomic, and metabolomic data is required.
Spatial Distribution and GBM Progression	Iba1 expression differs in different GBM phenotypes [41,70,123] and between different tissue areas of the same tumor.	Insufficient understanding of how spatial distribution of TAMs or Iba1+ cells can influence GBM progression.

**Table 2 cancers-17-01457-t002:** Disease conditions with known expression of CD163 by microglial cells.

Disease Conditions	Reference(s)
Alzheimer’s Disease	[138,139]
Glioblastoma	[140,141]
HIV encephalitis (HIVE)	[139]
Simian Immunodeficiency Virus Encephalitis (SIVE)	[135,137]
Traumatic Brain Injury	[142,143]
Variant Creutzfeldt-Jakob Disease	[139]

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
