# Peer review of "TAMing Gliomas: Unraveling the Roles of Iba1 and CD163 in Glioblastoma"

_cancers, 2025, doi:10.3390/cancers17091457_

Round 1
Reviewer 1 Report
Comments and Suggestions for Authors
The review describes how Iba1 and CD163 impacts glioblastoma. It is an interesting topic, and the manuscript could be improved based on the following comments/suggestions:
- No introduction?
- In general, the review topic flow/sequence is all over the place and may be confusing to readers. For example, “The tumour microenvironment in glioblastoma” is better tackled on the first part of the review with emphasis on microglia/TAM. Then, succeeding topic will be like describing a entire picture down to pixels.
- The review lacks authors’ scrutiny of published work. Authors can highlight controversies and debates regarding the topic, identify gaps on key findings from published work, etc. For example, if Iba1’s role is beyond just a marker, would targeting it potentially help control glioblastoma? Is Iba1 expressed on every TAM subset? If yes, what of the TAM that have potential anti-tumor activities should Iba1 is targeted.
- The review lacks thorough discussion on clinical trials involving microglia/TAMs in glioblastoma and how results from these trials may have been influenced by what is now known about Iba1 and CD163
- The authors should add final impact statements at end of every important paragraph before new subheading/section.
- The text in lines 45-74 does not reflect the subheading “The increasingly appreciated role of tumor-associated microglia/macrophages (TAMs) in glioblastoma,” as it mainly talked about origins and TAMs composition in glioblastoma microenvironment. Role was tackled superficially and more like an overview of said role.
Author Response
Comments: The review describes how Iba1 and CD163 impacts glioblastoma. It is an interesting topic, and the manuscript could be improved based on the following comments/suggestions:
- No introduction?
We have followed the referee’s suggestion and renamed the first section more aptly as ‘Introduction’.
- In general, the review topic flow/sequence is all over the place and may be confusing to readers. For example, is better tackled on the first part of the review with emphasis on microglia/TAM. Then, succeeding topic will be like describing a entire picture down to pixels.
We thank the referee for their suggestion. We have moved the section 'The tumour microenvironment in glioblastoma' to the front and we have also reorganized the sequence of the more general discussion sections at the end of the manuscript: 'TAMs undergo metabolic reprogramming', 'Role of microglia and macrophages in glioblastoma progression', and 'TAMs weaken the BBB supporting GBM invasion and growth'.
- The review lacks authors’ scrutiny of published work. Authors can highlight controversies and debates regarding the topic, identify gaps on key findings from published work, etc. For example, if Iba1’s role is beyond just a marker, would targeting it potentially help control glioblastoma? Is Iba1 expressed on every TAM subset? If yes, what of the TAM that have potential anti-tumor activities should Iba1 is targeted.
We have in fact addressed a number of knowledge gaps in Table 1, but perhaps this was not sufficiently clear. We have therefore extended the relevant sentence in the text that references the table. It now reads (lines 316-317): ‘The significance of Iba1 in GBM research is outlined in Table 1, which also identifies key areas where further investigation is needed to address existing knowledge gaps.'
The role of Iba1 as a potential means to help control glioblastoma growth is discussed in the section 'Iba1 as a potential future therapeutic target in glioblastoma', where experimental work using Iba1-siRNA is highlighted.
Iba1 is mainly expressed in microglia and some infiltrating macrophages, according to the current literature. Targeting Iba1 directly has so far not demonstrated anti-tumor activity. However, it has been shown that a reduction in the number of Iba1-expressing cells leads to better outcomes by shifting TAMs towards a pro-inflammatory M1-like phenotype.
- The review lacks thorough discussion on clinical trials involving microglia/TAMs in glioblastoma and how results from these trials may have been influenced by what is now known about Iba1 and CD163
The following information has been added to the paper at lines 647-667:
Therapeutic agents aimed at targeting TAMs in the TME to combat GBM progression have recently gained significant attention. M-CSF, a cytokine essential for TAM differentiation and GBM progression (210), has emerged as a promising target candidate for reducing GBM growth. Inhibiting M-CSF or its receptor, colony-stimulating factor-1 receptor (CSF-1R), has shown promise in preclinical GBM models (211). A small oral agent, PLX3397, was developed to directly target CSF-1R, demonstrating efficacy in GBM preclinical studies (212). However, human clinical trials (NCT01349036, NCT01790503) have re-ported limited efficacy (213,214). Another CSF-1R inhibitor, BLZ945, has been shown to improve survival rates (211) and enhance radiotherapy efficacy in glioma mouse models (215). In humans, BLZ945 can augment the efficacy of anti-PD-1 therapy against GBM by attenuating the immunosuppressive polarization of CD163+ macrophages (216). However, prolonged treatment with BLZ945 has yielded no additional benefits (217).
CD47, a "don't eat me" signal on tumor cells (218), has also been investigated as a potential target. An in vivo study found increased GBM phagocytosis following anti-CD47 treatment (219), whereas another model demonstrated minimal effects on glioma growth with anti-CD47 monotherapy (220). Moreover, anti-CD47 treatment can induce haematological toxicity (221). To date, human studies on anti-CD47 therapy are limited, and further research is needed to fully understand its potential. The TGF-βR1 inhibitor SB431542 has been shown to impair tumor growth by 75% in a mouse model (222), but the translatability of these findings to human studies remains uncertain (223,224).
References:
- Wu K, Lin K, Li X, Yuan X, Xu P, Ni P, et al. Redefining Tumor-Associated Macrophage Subpopulations and Func-tions in the Tumor Microenvironment. Front Immunol. 2020 Aug 4;11:1731.
- Pyonteck SM, Akkari L, Schuhmacher AJ, Bowman RL, Sevenich L, Quail DF, et al. CSF-1R inhibition alters mac-rophage polarization and blocks glioma progression. Nat Med. 2013 Oct;19(10):1264–72.
- Yan D, Kowal J, Akkari L, Schuhmacher AJ, Huse JT, West BL, et al. Inhibition of colony stimulating factor-1 recep-tor abrogates microenvironment-mediated therapeutic resistance in gliomas. Oncogene. 2017 Oct 26;36(43):6049–58.
- Butowski NA, Colman H, De Groot JF, Omuro AMP, Nayak L, Cloughesy TF, et al. A phase 2 study of orally ad-ministered PLX3397 in patients with recurrent glioblastoma. J Clin Oncol. 2014 May 20;32(15_suppl):2023–2023.
- Mendez JS, Cohen AL, Eckenstein M, Jensen RL, Burt LM, Salzman KL, et al. Phase 1b/2 study of orally adminis-tered pexidartinib in combination with radiation therapy and temozolomide in patients with newly diagnosed glioblastoma. Neuro-Oncol Adv. 2024 Nov 22;6(1):vdae202.
- ALMAHARIQ FM, QUINN JT, KESARWANI P, KANT S, MILLER CR, CHINNAIYAN P. Inhibition of Colony-Stimulating Factor-1 Receptor Enhances the Efficacy of Radiotherapy and Reduces Immune Suppression in Glio-blastoma. In Vivo. 2021 Jan 3;35(1):119–29.
- Cui X, Ma C, Vasudevaraja V, Serrano J, Tong J, Peng Y, et al. Dissecting the immunosuppressive tumor microenvi-ronments in Glioblastoma-on-a-Chip for optimized PD-1 immunotherapy. eLife. 9:e52253.
- Quail DF, Bowman RL, Akkari L, Quick ML, Schuhmacher AJ, Huse JT, et al. The tumor microenvironment under-lies acquired resistance to CSF1R inhibition in gliomas. Science. 2016 May 20;352(6288):aad3018.
- Willingham SB, Volkmer JP, Gentles AJ, Sahoo D, Dalerba P, Mitra SS, et al. The CD47-signal regulatory protein alpha (SIRPa) interaction is a therapeutic target for human solid tumors. Proc Natl Acad Sci U S A. 2012 Apr 24;109(17):6662–7.
- Zhang M, Hutter G, Kahn SA, Azad TD, Gholamin S, Xu CY, et al. Anti-CD47 Treatment Stimulates Phagocytosis of Glioblastoma by M1 and M2 Polarized Macrophages and Promotes M1 Polarized Macrophages In Vivo. PLOS ONE. 2016 Apr 19;11(4):e0153550.
- von Roemeling CA, Wang Y, Qie Y, Yuan H, Zhao H, Liu X, et al. Therapeutic modulation of phagocytosis in glio-blastoma can activate both innate and adaptive antitumour immunity. Nat Commun. 2020 Mar 20;11(1):1508.
- Sikic BI, Lakhani N, Patnaik A, Shah SA, Chandana SR, Rasco D, et al. First-in-Human, First-in-Class Phase I Trial of the Anti-CD47 Antibody Hu5F9-G4 in Patients With Advanced Cancers. J Clin Oncol. 2019 Apr 20;37(12):946–53.
- Liu H, Sun Y, Zhang Q, Jin W, Gordon RE, Zhang Y, et al. Pro-inflammatory and proliferative microglia drive pro-gression of glioblastoma. Cell Rep [Internet]. 2021 Sep 14 [cited 2025 Apr 17];36(11). Available from: https://www.cell.com/cell-reports/abstract/S2211-1247(21)01167-0
- Wick A, Desjardins A, Suarez C, Forsyth P, Gueorguieva I, Burkholder T, et al. Phase 1b/2a study of galunisertib, a small molecule inhibitor of transforming growth factor-beta receptor I, in combination with standard te-mozolomide-based radiochemotherapy in patients with newly diagnosed malignant glioma. Invest New Drugs. 2020;38(5):1570–9.
- Hollander MW den, Bensch F, Glaudemans AWJM, Munnink THO, Enting RH, Dunnen WFA den, et al. TGF-β Antibody Uptake in Recurrent High-Grade Glioma Imaged with 89Zr-Fresolimumab PET. J Nucl Med. 2015 Sep 1;56(9):1310–4.
4)The authors should add final impact statements at end of every important paragraph before new subheading/section.
Impact statements have been inserted at the following lines:124-128, 211-212, 292-294.
- The text in lines 45-74 does not reflect the subheading “The increasingly appreciated role of tumor-associated microglia/macrophages (TAMs) in glioblastoma,” as it mainly talked about origins and TAMs composition in glioblastoma microenvironment. Role was tackled superficially and more like an overview of said role.
We agree with the reviewer and have revised this section accordingly. The section is now titled 'Introduction', and the former heading has been incorporated into the text as a statement.
Reviewer 2 Report
Comments and Suggestions for Authors
The manuscript could not be considered for publication in the current form for following reasons.
Major Comments:
- Authors has stated that “We are particularly interested in microglia, the resident brain macrophage precursors that normally express Iba1 but not CD163”…Recent studies show CD163 can be expressed by activated microglia in gliomas. I request authors to consider explaining the binary classification according to the plasticity, especially given CD163 upregulation in tumor microglia versus other disease conditions.
- The authors mentioned necrotic areas attract CD163+ macrophages in the introduction part, but evidence is missing: I recommend the authors to provide some relevant updated literature references, especially for single cell data evidencing TAMs vs. microglia along with Iba and CD163 localization in necrosis.
- Authors stated that"Iba1 is widely used as a marker for brain resident microglia” while recent studies detail Iba1 is also expressed in infiltrating. How would authors like to comment on this?
- The authors mentioned Iba1's role in cytoskeletal reorganization but the functional mechanisms are not clearly discussed. I request authors to discuss its requirement in synapse formation, and impact of impaired synaptic engulfment in iba1 KO models.
- Authors have mentioned the dual role of iba1, I suggest them to mention the importance of proinflammatory function.
- The authors have noticed an increase in Iba1 in activated microglia, but there are multiple previous publications stating that Iba1 may label both resting and activated state, how would authors like to address this?
- The authors discussed that Iba1-siRNA reduces microglial migration, but there is a lack of information concerning in vivo GBM models showing tumor regression. I request that authors to provide some information with updated references.
- The authors claimed that: “CD163+ cells dominate necrotic areas while Iba1+ cells associate with viable tumor regions,” but it is contradicting with existing literature; please verify and try to modify. (Nature 2024 s41698-024-00692-w, PMC8284623).
- The authors mentioned that: CD163 is "exclusively expressed in macrophage lineages" but Recent scRNA-seq data revealed that microglial CD163 in recurrent GBM (JCI 2023). I request authors to consider this and rewrite.
Minor Comments:
- Can consider including the clinical correlation of CD163 prognostic value in wild-type GBM.
- Update the recent literature references.
Author Response
Comments: The manuscript could not be considered for publication in the current form for following reasons.
Major Comments:
- Authors has stated that “We are particularly interested in microglia, the resident brain macrophage precursors that normally express Iba1 but not CD163”…Recent studies show CD163 can be expressed by activated microglia in gliomas. I request authors to consider explaining the binary classification according to the plasticity, especially given CD163 upregulation in tumor microglia versus other disease conditions.
There is no contradiction. We agree with the reviewer that 'CD163 can be expressed by activated microglia in gliomas.' CD163 is not normally expressed in CNS microglia, but its expression by microglia can be observed in certain disease conditions, including glioma (cf. Table 2 and Figure 4H in one of our own recent publications, Loh et al.). However, it is worth noting that the number of microglial cells involved appears to differ significantly across diseases, and this also seems to hold true for the rodent homologue of CD163, ED2. According to Woolf et al.[1], there is an increased expression of CD163 in tumor microglia compared to microglia in epileptic brain. The proposed, yet not well-understood, explanation behind this phenomenon is based on the long-debated binary model of TAM activation states (M1 vs M2), where upregulation of CD163 is thought to be associated with a shift towards an immunosuppressive M2 state [2,3]. In epilepsy, there is no shift from a surveillance M1 state to an M2 state; therefore, CD163 is not upregulated. However, this binary classification is controversial, as it oversimplifies the activation of TAMs in the glioma context.
References
1: Zoe Woolf, Molly E V Swanson, Leon C Smyth, Edward W Mee, Patrick Schweder, Peter Heppner, Bernard J H Kim, Clinton Turner, Robyn L Oldfield, Maurice A Curtis, Richard L M Faull, Emma L Scotter, Thomas I-H Park, Michael Dragunow, Single-cell image analysis reveals a protective role for microglia in glioblastoma, Neuro-Oncology Advances, Volume 3, Issue 1, January-December 2021, vdab031, https://doi.org/10.1093/noajnl/vdab031)
2: Solomou G, Young AMH, Bulstrode HJCJ. Microglia and macrophages in glioblastoma: landscapes and treatment directions. Mol Oncol. 2024 Dec;18(12):2906-2926. doi: 10.1002/1878-0261.13657. Epub 2024 May 7. PMID: 38712663; PMCID: PMC11619806.
3: Miller, T. E., El Farran, C. A., Couturier, C. P., Chen, Z., D'Antonio, J. P., Verga, J., Villanueva, M. A., Castro, L. N. G., Tong, Y. E., Saadi, T. A., Chiocca, A. N., Fischer, D. S., Heiland, D. H., Guerriero, J. L., Petrecca, K., Suva, M. L., Shalek, A. K., & Bernstein, B. E. (2023). Programs, Origins, and Niches of Immunomodulatory Myeloid Cells in Gliomas. bioRxiv : the preprint server for biology, 2023.10.24.563466. https://doi.org/10.1101/2023.10.24.563466
- The authors mentioned necrotic areas attract CD163+ macrophages in the introduction part, but evidence is missing: I recommend the authors to provide some relevant updated literature references, especially for single cell data evidencing TAMs vs. microglia along with Iba and CD163 localization in necrosis.
We have changed lines 417-424 as follows:
Single-cell transcriptomic studies have identified distinct markers for microglia (P2RY12 and TMEM119) and TAMs (CD14 and CD163), although there is some overlap in marker expression (72). In glioblastoma, Iba1-positive cells (including both microglia and macrophages) are often found demarcating necrotic areas. In contrast, CD163 can show extensive extracellular deposition within necrotic tumor areas(167). The differential localization of these markers in relation to necrosis suggests different roles for TAMs and microglia (70).
References:
- Loh C, Zheng Y, Alzoubi I, Alexander KL, Lee M, Cai WD, et al. Microglia and brain macrophages are differentially associated with tumor necrosis in glioblastoma: A link to tumor progression. Oncol Res. 2024;0(0):1–10.
- Woolf Z, Swanson MEV, Smyth LC, Mee EW, Schweder P, Heppner P, et al. Single-cell image analysis reveals a protective role for microglia in glioblastoma. Neuro-Oncol Adv. 2021;3(1):vdab031.
- Tamma R, Ingravallo G, Annese T, d’Amati A, Lorusso L, Ribatti D. Tumor Microenvironment and Microvascular Density in Human Glioblastoma. Cells. 2023 Jan;12(1):11.
- Authors stated that"Iba1 is widely used as a marker for brain resident microglia” while recent studies detail Iba1 is also expressed in infiltrating. How would authors like to comment on this?
We have changed the statement as follows:
Iba1 is now commonly used as a marker for brain-resident microglia, although recent studies have shown that it can also be expressed by infiltrating macrophages (40–43). This highlights the need for careful consideration of immune cell heterogeneity in the brain under pathological conditions.
References:
- Roesch S, Rapp C, Dettling S, Herold-Mende C. When Immune Cells Turn Bad—Tumor-Associated Micro-glia/Macrophages in Glioma. Int J Mol Sci. 2018 Feb;19(2):436.
- Kaffes I, Szulzewsky F, Chen Z, Herting CJ, Gabanic B, Vega JEV. Human Mesenchymal glioblastomas are charac-terized by an increased immune cell presence compared to Proneural and Classical tumors. Oncoimmunology. 2019 Aug;22;8(11):e1655360.
- Liesche-Starnecker F, Mayer K, Kofler F, Baur S, Schmidt-Graf F, Kempter J. Immunohistochemically Characterized Intratumoral Heterogeneity Is a Prognostic Marker in Human Glioblastoma. Cancers. 2020 Oct;13;12(10):2964.
- Annovazzi L, Mellai M, Bovio E, Mazzetti S, Pollo B, Schiffer D. Microglia immunophenotyping in gliomas. Oncol Lett. 2017 Nov 9;15(1):998.
- The authors mentioned Iba1's role in cytoskeletal reorganization but the functional mechanisms are not clearly discussed. I request authors to discuss its requirement in synapse formation, and impact of impaired synaptic engulfment in iba1 KO models.
We have added the following paragraph:
Recent work shows that Iba1 is also involved in regulating synaptic development and function. Specifically, Iba1 plays a crucial role in the formation of excitatory synapses in the juvenile brain (48). Studies using Iba1-deficient mice have demonstrated that Iba1 promotes synapse formation rather than limiting synaptic pruning. In Iba1-knockout mice (48) a reduced number of excitatory synapses results in changes in behavior, including impairments in object recognition memory and social interaction. Interestingly, Iba1 knockout models exhibit diminished microglial synaptic engulfment capacity, which may be a compensatory response to the early deficit in synapse formation. Synapse formation requires the extension or retraction of processes, which is accomplished by Iba1-mediated microglial protrusions that communicate with the surrounding environment. This pro-cess is thought to involve cytoskeletal reorganization within microglia, mainly via the dynamics of actin filaments (49). The importance of Iba1 is illustrated in the formation and remodeling of parallel actin bundles, a crucial scaffold structure for motility protrusions such as lamellipodia and filopodia (49,50). The mechanism behind microglial synapse engulfment, or synaptic pruning, involves Iba1 contributing to the formation of membrane ruffles, followed by the formation of a phagocytic cup through actin bundling (46). This al-lows microglia to phagocytose and clean up unwanted synapses.
References:
- De Leon-Oliva D, Garcia-Montero C, Fraile-Martinez O, Boaru DL, García-Puente L, Rios-Parra A, et al. AIF1: Func-tion and Connection with Inflammatory Diseases. Biology. 2023 May 9;12(5):694.
- Lituma PJ, Woo E, O’Hara BF, Castillo PE, Sibinga NES, Nandi S. Altered synaptic connectivity and brain function in mice lacking microglial adapter protein Iba1. Proc Natl Acad Sci U S A. 2021 Nov 16;118(46):e2115539118.
- Franco-Bocanegra DK, McAuley C, Nicoll JAR, Boche D. Molecular Mechanisms of Microglial Motility: Changes in Ageing and Alzheimer’s Disease. Cells. 2019 Jun 25;8(6):639.
- Blanchoin L, Boujemaa-Paterski R, Sykes C, Plastino J. Actin dynamics, architecture, and mechanics in cell motili-ty. Physiol Rev. 2014 Jan;94(1):235–63.
- Authors have mentioned the dual role of iba1, I suggest them to mention the importance of proinflammatory function.
We have added the following paragraph:
Moreover, Iba1 plays a significant role in proinflammatory processes in the CNS (66–68). Upon exposure to inflammatory stimuli, such as IFN-γ, Iba1 is upregulated, leading to microglial activation (56,57). This upregulation promotes the secretion of proinflammatory cytokines and chemokines, including IL-6, thereby enhancing inflammatory responses (60). At the molecular level, Iba1 interacts with the actin cytoskeleton and activates Rac GTPase signalling in microglia, which is crucial for cell motility and phagocytosis, highlighting its role in inflammation (44,45). Furthermore, Iba1 expression is upregulated in response to ischemic injury, suggesting its participation in the sterile inflammatory response that follows ischemia, and potentially contributing to the ensuing pathophysiological consequences of ischemia (69).
References:
- Kanazawa H, Ohsawa K, Sasaki Y, Kohsaka S, Imai Y. Macrophage/microglia-specific protein Iba1 enhances membrane ruffling and Rac activation via phospholipase C-gamma -dependent pathway. J Biol Chem. 2002 May 31;277(22):20026–32.
- Ohsawa K, Imai Y, Kanazawa H, Sasaki Y, Kohsaka S. Involvement of Iba1 in membrane ruffling and phagocytosis of macrophages/microglia. J Cell Sci. 2000 Sep;113 ( Pt 17):3073–84.
- Morohashi T, Iwabuchi K, Watano K, Dashtsoodol N, Mishima T, Nakai Y, et al. Allograft inflammatory factor-1 regulates trinitrobenzene sulphonic acid-induced colitis. Immunology. 2003 Sep;110(1):112–9.
- Kelemen SE, Autieri MV. Expression of Allograft Inflammatory Factor-1 in T Lymphocytes. Am J Pathol. 2005 Aug;167(2):619–26.
- Watano K, Iwabuchi K, Fujii S, Ishimori N, Mitsuhashi S, Ato M, et al. Allograft inflammatory factor-1 augments production of interleukin-6, -10 and -12 by a mouse macrophage line. Immunology. 2001 Nov;104(3):307–16.
- Chinnasamy P, Lutz SE, Riascos-Bernal DF, Jeganathan V, Casimiro I, Brosnan CF, et al. Loss of Allograft Inflammatory Factor-1 Ameliorates Experimental Autoimmune Encephalomyelitis by Limiting Encephalitogenic CD4 T-Cell Expansion. Mol Med Camb Mass. 2015 Jan 6;21(1):233–41.
- Oliveira B da S, Toscano EC de B, Abreu LKS, Fernandes H de B, Amorim RF, Ferreira RN, et al. Nigrostriatal Inflammation Is Associated with Nonmotor Symptoms in an Experimental Model of Prodromal Parkinson’s Disease. Neuroscience. 2024 Jun 21;549:65–75.
- Hopperton KE, Mohammad D, Trépanier MO, Giuliano V, Bazinet RP. Markers of microglia in post-mortem brain samples from patients with Alzheimer’s disease: a systematic review. Mol Psychiatry. 2018 Feb;23(2):177–98.
- Luheshi NM, Kovács KJ, Lopez-Castejon G, Brough D, Denes A. Interleukin-1α expression precedes IL-1β after ischemic brain injury and is localised to areas of focal neuronal loss and penumbral tissues. J Neuroinflammation. 2011 Dec 29;8:186.
- The authors have noticed an increase in Iba1 in activated microglia, but there are multiple previous publications stating that Iba1 may label both resting and activated state, how would authors like to address this?
We have added the following sentence:
Notably, Iba1 can label both resting and activated microglia. However, our results align with other studies (63–65) that demonstrate a significant increase in Iba1 expression in microglia responding to glioma. This upregulation is accompanied by a deramification of microglia, leading to a more rounded, macrophage-like morphology, which typically exhibits a gradual transition forming a tissue gradient, as illustrated in Figure 1.
References:
- Zheng Y, Fuse H, Alzoubi I, Graeber MB. Microglia-Derived Brain Macrophages Associate with Glioblastoma Stem Cells: A Potential Mechanism for Tumor Progression Revealed by AI-Assisted Analysis. Cells. 2025 Jan;14(6):413.
- Kvisten M, Mikkelsen VE, Stensjøen AL, Solheim O, Van Der Want J, Torp SH. Microglia and macrophages in hu-man glioblastomas: A morphological and immunohistochemical study. Mol Clin Oncol. 2019 Jul;11(1):31–6.
- Lier J, Streit WJ, Bechmann I. Beyond Activation: Characterizing Microglial Functional Phenotypes. Cells. 2021 Aug 28;10(9):2236
- The authors discussed that Iba1-siRNA reduces microglial migration, but there is a lack of information concerning in vivo GBM models showing tumor regression. I request that authors to provide some information with updated references.
No in vivo GBM models using Iba1-siRNA so far shows tumor regression.
- The authors claimed that: “CD163+ cells dominate necrotic areas while Iba1+ cells associate with viable tumor regions,” but it is contradicting with existing literature; please verify and try to modify. (Nature 2024 s41698-024-00692-w, PMC8284623).
The authors of PMC8284623 state that "necrotic areas were avoided during analysis". Furthermore, s41698-024-00692-w does not mention the term "necrosis". Therefore, we are unable to address this point. This may be due to a misunderstanding, and we would appreciate clarification on this matter.
- The authors mentioned that: CD163 is "exclusively expressed in macrophage lineages" but Recent scRNA-seq data revealed that microglial CD163 in recurrent GBM (JCI 2023). I request authors to consider this and rewrite.
Microglia are indeed considered to belong to the macrophage lineage.
Minor Comments:
- Can consider including the clinical correlation of CD163 prognostic value in wild-type GBM.
CD163 has been suggested to be a promising biomarker and therapeutic target for GBM.
Reference:
Liu S, Zhang C, Maimela NR, Yang L, Zhang Z, Ping Y, Huang L, Zhang Y. Molecular and clinical characterization of CD163 expression via large-scale analysis in glioma. Oncoimmunology. 2019 Apr 17;8(7):1601478. doi: 10.1080/2162402X.2019.1601478. PMID: 31143523; PMCID: PMC6527268.
This information has been added.
- Update the recent literature references.
The most recent literature references have been updated.
Reviewer 3 Report
Comments and Suggestions for Authors
The manuscript “TAMing Gliomas: Unravelling the roles of Iba1 and CD163 in Glioblastoma “ is about the role of tumor-associated macrophages in glioma growth. This area is very important and many papers were published where the tumor microenvironment is discussed in connection to glioblastoma progression. The authors paid the main attention to two proteins, Iba1 and CD163. The ionized calcium-binding adapter molecule 1 (Iba1) or AIF1 (allograft inflammatory factor 1) is widely used as a marker for brain resident microglia. It is involved in intracellular signaling, cytoskeletal reorganization, and phagocytosis. Another protein, CD163 is the high affinity scavenger receptor and a marker of macrophage/monocyte lineage cells. The authors applied the program PathoFusion, an artificial intelligence (AI) framework, to check the localization of these proteins and morphological variations in the tumor microenvironment. Thus, this approach could allow us to find different patterns in microglial cell populations. The manuscript will be interesting and useful for the scientists involved in the study on glioblastoma, but there are some points that need to be corrected.
Recommendations.
- The name Iba1 (the ionized calcium-binding adapter molecule 1) is an alternative name for AIF1 (allograft inflammatory factor 1). The preferred name is AIF1, and under this name, the protein is present in all databases. The section “Overview of Iba1 structure, function and its expression in normal and pathologically altered brain tissue” should be expanded according to information about AIF1.
- The section “Role of Iba1 in glioblastoma progression” is very speculative and should be rewritten. Figure 2. Pathways proposed to be involved in Iba1 effects on GBM progression. – The Figure is based on a very shaky assumption. Actually, there is no evidence that Iba1 is involved in NHE1-initiated pathways. The concurrent expression is not a proof.
- A considerable attention is paid to TGF-β, but there are no solid proofs of association of AIF1with TGF-β activation. Actually, the authors themselves mention an inhibitory relationship between Iba1 and TGF-β as well.
The minors
- “…it appears reasonable to associate Iba1+ with TGF-β production.” This phrase looks clumsy. A word microglia is missing?
- The list of abbreviations is missing.
- In many cases, the space is missing. For instance, “…macrophages (TAMs)(1–3). Should be macrophages (TAMs) (1–3).
- Capitalize the word “roles” appropriately in the title.
Author Response
Comments: The manuscript “TAMing Gliomas: Unravelling the roles of Iba1 and CD163 in Glioblastoma “ is about the role of tumor-associated macrophages in glioma growth. This area is very important and many papers were published where the tumor microenvironment is discussed in connection to glioblastoma progression. The authors paid the main attention to two proteins, Iba1 and CD163. The ionized calcium-binding adapter molecule 1 (Iba1) or AIF1 (allograft inflammatory factor 1) is widely used as a marker for brain resident microglia. It is involved in intracellular signaling, cytoskeletal reorganization, and phagocytosis. Another protein, CD163 is the high affinity scavenger receptor and a marker of macrophage/monocyte lineage cells. The authors applied the program PathoFusion, an artificial intelligence (AI) framework, to check the localization of these proteins and morphological variations in the tumor microenvironment. Thus, this approach could allow us to find different patterns in microglial cell populations. The manuscript will be interesting and useful for the scientists involved in the study on glioblastoma, but there are some points that need to be corrected.
Recommendations.
- The name Iba1 (the ionized calcium-binding adapter molecule 1) is an alternative name for AIF1 (allograft inflammatory factor 1). The preferred name is AIF1, and under this name, the protein is present in all databases. The section “Overview of Iba1 structure, function and its expression in normal and pathologically altered brain tissue” should be expanded according to information about AIF1.
The following text has been added at lines 161-182
Iba1, also known as Allograft Inflammatory Factor 1 (AIF1), has emerged as a useful biomarker of immune activation (51). Its role in cardiac allograft rejection was first highlighted by Ustans et al. (52), who observed that AIF1 was consistently expressed in chronically rejected cardiac allografts but absent in syngeneic grafts, underscoring its potential as a marker that promotes transplant rejection. This observation was later extended to kidney transplant models (53–55), where AIF1 was similarly associated with allograft rejection. However, one study (54) reported no such correlation, suggesting that its role may vary depending on the context or model used. Beyond transplantation, AIF1 has been implicated in a wide range of inflammatory and immune-related diseases, including rheumatoid arthritis, atherosclerosis, certain CNS disorders, and metabolic syndromes, as reviewed by Sikora et al. (51). Expression of AIF1 is induced by inflammatory cytokines, including interferon-gamma (IFN-γ), and contributes to the modulation of immune responses, particularly by influencing the function of T helper 1 (Th1) cells, as evidenced by studies in colitis mouse models (56,57).
In the context of CNS disorders, AIF1 has also been recognized as a marker of microglial and monocyte activation, particularly in meningoencephalitis models (58,59), emphasizing its significance in immune modulation within the brain. Notably, in RAW 264.7, a macrophage cell line, when transfected with AIF1 cDNA, AIF1 is overexpressed, followed by a significant increase in the levels of IL-6, IL-10, and IL-12p40 upon bacterial lipopolysaccharide (LPS) stimulation (60). These results suggest that AIF1 is not only a marker for macrophages and microglia but also supports the role of macrophages in immune responses.
References:
- Sikora M, Kopeć B, Piotrowska K, Pawlik A. Role of allograft inflammatory factor-1 in pathogenesis of diseases. Immunol Lett. 2020 Feb;218:1–4.
- Utans U, Arceci RJ, Yamashita Y, Russell ME. Cloning and characterization of allograft inflammatory factor-1: a novel macrophage factor identified in rat cardiac allografts with chronic rejection. J Clin Invest. 1995 Jun 1;95(6):2954–62.
- McDaniel DO, Rigney DA, McDaniel KY, Windham WJ, Redmond P, Williams B, et al. Early Expression Profile of Inflammatory Markers and Kidney Allograft Status. Transplant Proc. 2013 May 1;45(4):1520–3.
- Romanowski M, KÅ‚oda K, Milczarek S, Pawlik A, DomaÅ„ski L, Safranow K, et al. Influence of AIF1 Gene Polymor-phisms on Long-Term Kidney Allograft Function. Ann Transplant. 2015 Sep 1;20:506–11.
- Vu D, Tellez-Corrales E, Shah T, Hutchinson I, Min DI. Influence of Cyclooxygenase-2 (COX-2) gene promoter-1195 and allograft inflammatory factor-1 (AIF-1) polymorphisms on allograft outcome in Hispanic kidney transplant recipients. Hum Immunol. 2013 Oct 1;74(10):1386–91.
- Morohashi T, Iwabuchi K, Watano K, Dashtsoodol N, Mishima T, Nakai Y, et al. Allograft inflammatory factor-1 regulates trinitrobenzene sulphonic acid-induced colitis. Immunology. 2003 Sep;110(1):112–9.
- Kelemen SE, Autieri MV. Expression of Allograft Inflammatory Factor-1 in T Lymphocytes. Am J Pathol. 2005 Aug;167(2):619–26.
- Herden C, Schluesener HJ, Richt JA. Expression of allograft inflammatory factor-1 and haeme oxygenase-1 in brains of rats infected with the neurotropic Borna disease virus. Neuropathol Appl Neurobiol. 2005 Oct;31(5):512–21.
- Deininger MH, Weinschenk T, Meyermann R, Schluesener HJ. The allograft inflammatory factor-1 in Creutzfeldt-Jakob disease brains. Neuropathol Appl Neurobiol. 2003 Aug;29(4):389–99.
- Watano K, Iwabuchi K, Fujii S, Ishimori N, Mitsuhashi S, Ato M, et al. Allograft inflammatory factor-1 augments production of interleukin-6, -10 and -12 by a mouse macrophage line. Immunology. 2001 Nov;104(3):307–16.
- The section “Role of Iba1 in glioblastoma progression” is very speculative and should be rewritten. Figure 2. Pathways proposed to be involved in Iba1 effects on GBM progression. – The Figure is based on a very shaky assumption. Actually, there is no evidence that Iba1 is involved in NHE1-initiated pathways. The concurrent expression is not a proof.
We thank the referee for their critical comments. We have modified this section and Figure 2 accordingly. Highly speculative statements have been removed and the text has been much shortened as a result. The figure has been modified accordingly and speculation is clearly marked as such.
The following text has been transferred to lines 215-237:
The involvement of Iba1/Aif1 molecules in glioma progression, particularly in glioblastoma, is supported by accumulating evidence. For instance, Iba1+ microglia have been found to associate more closely with GSCs than CD163+ macrophages (63). Furthermore, high levels of Iba1 expression are correlated with reduced patient survival, suggesting its potential as a prognostic marker (71). Notably, in our view this association is not contradicted by the study of Woolf et al. (72), which had a limited sample size. Microglia, marked by Iba1, contribute to the creation of an immunosuppressive environment that supports tumor growth. Microglia and other macrophages can exert pro-tumorigenic effects through the production of anti-inflammatory cytokines, such as transforming growth factor-β (TGF-β), which suppresses the normal function of Th1 cells (73). A retrospective data analysis of 1,270 glioma patients also demonstrated a potential correlation between Iba1 expression and GBM tumorigenesis (74). TGF-β has been shown to promote GBM cell proliferation, invasion, angiogenesis and immunosuppression (75). The association be-tween Iba1 and TGF-β is complex, with some evidence suggesting an inhibitory relationship (76). Silencing of Iba1 in an in-vivo model demonstrated a significant increase in IL-10 levels in T cells under inflammatory tissue condition (77). Moreover, M2-polarised microglia have been shown to upregulate angiogenesis in the GBM microenvironment via supporting the transport of circKIF18A to human brain microvessel endothelial cells (hBMECs) (78). In fact, microglia appear to be more important than peripheral macrophages in promoting angiogenesis (79). Reducing the number of microglial cells had a significant negative impact on the formation of tumoral blood vessels (79). Figure 2 shows Iba1 related pathways that are potentially involved in GBM progression.
References:
- Zheng Y, Fuse H, Alzoubi I, Graeber MB. Microglia-Derived Brain Macrophages Associate with Glioblastoma Stem Cells: A Potential Mechanism for Tumor Progression Revealed by AI-Assisted Analysis. Cells. 2025 Jan;14(6):413.
- Liu X, Zhang D, Hu J, Xu S, Xu C, Shen Y. Allograft inflammatory factor 1 is a potential diagnostic, immunological, and prognostic biomarker in pan-cancer. Aging. 2023 Apr 3;15(7):2582.
- Woolf Z, Swanson MEV, Smyth LC, Mee EW, Schweder P, Heppner P, et al. Single-cell image analysis reveals a protective role for microglia in glioblastoma. Neuro-Oncol Adv. 2021;3(1):vdab031.
- Bouhlel MA, Derudas B, Rigamonti E, Dièvart R, Brozek J, Haulon S, et al. PPARgamma activation primes human monocytes into alternative M2 macrophages with anti-inflammatory properties. Cell Metab. 2007 Aug;6(2):137–43.
- Rao M, Yang Z, Huang K, Liu W, Chai Y. Correlation of AIF-1 Expression with Immune and Clinical Features in 1270 Glioma Samples. J Mol Neurosci MN. 2022 Feb;72(2):420–32.
- Wesolowska A, Kwiatkowska A, Slomnicki L, Dembinski M, Master A, Sliwa M, et al. Microglia-derived TGF-beta as an important regulator of glioblastoma invasion--an inhibition of TGF-beta-dependent effects by shRNA against human TGF-beta type II receptor. Oncogene. 2008 Feb 7;27(7):918–30.
- Cano-Martínez D, Monserrat J, Hernández-Breijo B, Sanmartín Salinas P, Álvarez-Mon M, Val Toledo-Lobo M, et al. Extracellular allograft inflammatory factor-1 (AIF-1) potentiates Th1 cell differentiation and inhibits Treg response in human peripheral blood mononuclear cells from normal subjects. Hum Immunol. 2020 Feb 1;81(2):91–100.
- Elizondo DM, Andargie TE, Yang D, Kacsinta AD, Lipscomb MW. Inhibition of Allograft Inflammatory Factor-1 in Dendritic Cells Restrains CD4+ T Cell Effector Responses and Induces CD25+Foxp3+ T Regulatory Subsets. Front Immunol [Internet]. 2017 Nov 8 [cited 2024 Oct 10];8. Available from: https://www.frontiersin.org/journals/immunology/articles/10.3389/fimmu.2017.01502/full
- Jiang Y, Zhao J, Xu J, Zhang H, Zhou J, Li H, et al. Glioblastoma-associated microglia-derived exosomal circKIF18A promotes angiogenesis by targeting FOXC2. Oncogene. 2022 Jun;41(26):3461–73.
- Brandenburg S, Müller A, Turkowski K, Radev YT, Rot S, Schmidt C, et al. Resident microglia rather than peripheral macrophages promote vascularization in brain tumors and are source of alternative pro-angiogenic factors. Acta Neuropathol (Berl). 2016 Mar;131(3):365–78.
- A considerable attention is paid to TGF-β, but there are no solid proofs of association of AIF1 with TGF-β activation. Actually, the authors themselves mention an inhibitory relationship between Iba1 and TGF-β as well.
An inhibitory relationship between Iba1 and TGF-β has indeed been observed in a study on peripheral blood which we have cited (1). However, this differs significantly from neuronal studies. The unique GBM microenvironment appears to reprogram the relationship between Iba1 and TGF-β, thereby promoting tumor progression.
This above is now stated in the text.
Reference:
1: Cano-Martínez D, Monserrat J, Hernández-Breijo B, Sanmartín Salinas P, Álvarez-Mon M, Val Toledo-Lobo M, et al. Extracellular allograft inflammatory factor-1 (AIF-1) potentiates Th1 cell differentiation and inhibits Treg response in human peripheral blood mononuclear cells from normal subjects. Hum Immunol. 2020 Feb 1;81(2):91–100.
The minors
- “…it appears reasonable to associate Iba1+ with TGF-β production.” This phrase looks clumsy. A word microglia is missing?
This change has been made.
- The list of abbreviations is missing.
Abbreviations
The following abbreviations are used in this manuscript:
- ADAM17
- Metalloproteinase 17
- AKT
- Protein kinase B
- AI
- Artificial intelligence
- BBB
- Blood brain barrier
- BCNN
- Bifocal convolutional neural network
- BMDMs
- Bone marrow-derived macrophages
- BTB
- Blood-tumor barrier
- CAR-T
- Chimeric antigen receptor T-cell
- CCL2
- C-C motif chemokine ligand 2
- CCL5
- C-C motif chemokine ligand 5
- CCR2
- C-C motif chemokine receptor 2
- circRNAs
- Exosomal circular RNAs
- ctDNA
- Circulating tumor DNA
- CNS
- Central nervous system
- CSF
- Cerebrospinal fluid
- CSF-1R
- Colony stimulating factor-1 receptor
- CXCL2
- CXC motif chemokine ligand 2
- DCs
- Dendritic cells
- ECM
- Extracellular matrix
- EGFR
- Epidermal growth factor receptor
- EMT
- Epithelial-mesenchymal transition
- ERK
- Extracellular signal-regulated kinase
- FOXC2
- Forkhead box protein 2
- GBM
- Glioblastoma
- GBM-hPMNL
- GBM-associated polymorphonuclear leukocytes/granulocytes
- GSC
- Glioma stem cell
- H&E
- Haematoxylin and eosin
- hBMECs
- human brain microvessel endothelial cells
- HIF
- Hypoxia inducible factor
- HIVE
- Human immunodeficiency virus encephalitis
- Iba1
- Ionized calcium-binding adapter molecule 1
- Iba-siRNA
- Iba1 small interfering RNA
- IFN-γ
- Interferon-γ
- IGFBP1
- Insulin-like growth factor-binding protein 1
- IHC
- Immunohistochemistry / immunohistochemical
- IL
- Interleukin
- LGGs
- Low-grade gliomas
- LPS
- Lipopolysaccharide
- MAPK
- Mitogen-activated protein kinase
- M-CSF
- Macrophage colony-stimulating factor
- MDSCs
- Myeloid-derived suppressor cells
- MEK
- Mitogen-activated protein kinase kinase
- MHC
- Major histocompatibility complex
- MMP14
- Matrix metalloproteinase 14
- MMP2
- Matrix metalloproteinase 2
- MMP9
- Matrix metalloproteinase 9
- MB
- Molecular barcoding
- mTOR
- Mechanistic target of rapamycin
- NFκB
- Nuclear factor kappa beta
- NHE1
- Sodium-hydrogen exchanger 1
- PDGF-C
- Platelet-derived growth factor-C
- PD-L1
- Programmed death-ligand 1
- PHD2
- Prolyl-4-hydroxylase 2
- PI3K
- Phosphoinositide 3-kinase
- PLC- ?
- Phospholipase C-gamma
- PSGL-1
- P-selectin glycoprotein ligand-1
- PTN
- Pleiotrophin
- PTPRZ1
- Protein tyrosine phosphatase receptor type Z1
- PVMs
- Perivascular macrophages
- Rac
- Ras-related C3 botulinum toxin substrate
- sCD163
- Surface CD163
- SCN3B
- Sodium channel β3 subunit
- scRNA-Seq
- Single-cell RNA sequencing
- SELP
- P-selectin
- SIVE
- Simian immunodeficiency virus encephalitis
- SMAD
- Mothers against decapentaplegic homolog
- Spp1
- Secreted phosphoprotein 1
- SRCR
- Scavenger receptor cysteine-rich domain
- TACE
- TNF-alpha converting enzyme
- TAM
- Tumor-associated macrophages
- TGF- β
- Transforming growth factor-beta
- Th1
- T helper type 1
- TIMP
- Tissue inhibitor of metalloproteinases
- TLR4
- Toll-like receptor 4
- TME
- Tumor microenvironment
- TNF-α
- Tumor necrosis factor-alpha
- TSP1
- Thrombospondin 1
- TWEAK
- TNF-like weak inducer of apoptosis
- VEGF
- Vascular endothelial growth factor
- WSI
- Whole-slide images
- In many cases, the space is missing. For instance, “…macrophages (TAMs)(1–3). Should be macrophages (TAMs) (1–3).
This change has been made.
- Capitalize the word “roles” appropriately in the title.
This change has been made.
Reviewer 4 Report
Comments and Suggestions for Authors
The manuscript presents a comprehensive review of the roles of tumor-associated macrophages in glioblastoma, focusing on the markers Iba1 and CD163. The authors examine the different sources of TAMs, their activation, and their implications in glioma growth and treatment. The integration of artificial intelligence for macrophage identification in tissue samples is a particularly noteworthy advancement.
Comments:
- The review could be enhanced by providing more detailed explanations of the immunomodulatory mechanisms by which Iba1 and CD163 exert their effects on gliomas
- Incorporate a dedicated section summarizing pertinent clinical data, particularly studies that assess the outcome of therapies targeting TAMs and their relationship with Iba1 and CD163 expression. This addition could enhance the clinical relevance of the manuscript significantly.
- The association of Iba1 with signaling pathways such as TGF-β is particularly insightful. The review highlights how TGF-β promotes immunosuppression and contributes to gliomagenesis, thereby implicating Iba1's involvement in critical tumor progression mechanisms. However, there could be a more thorough exploration of how Iba1 interacts with other signaling pathways and its direct effects on tumor cell behavior and immune evasion
- The manuscript should emphasize the therapeutic implications of targeting Iba1. Given its role in glioblastoma progression, there is a clear opportunity for developing therapies that modulate Iba1’s activity or its associated pathways. The discussion of preclinical data supporting this concept is valuable, but further elaboration on potential therapeutic modalities or ongoing clinical trials would strengthen this section
- The review mentions Iba1’s involvement in angiogenesis through its association with pro-angiogenic factors. Expanding this discussion could provide deeper insights into how increased Iba1 expression facilitates neovascularization within tumors.
- The discussion surrounding CD163's role in antigen presentation and its regulation in response to pro-inflammatory and anti-inflammatory signals is crucial. CD163, as a marker for TAMs, is pivotal in understanding the immune landscape of the tumor microenvironment in glioblastoma, particularly how these cells may facilitate or hinder anti-tumor immunity
- A list of abbreviations should be added to the manuscript
Author Response
Comments: The manuscript presents a comprehensive review of the roles of tumor-associated macrophages in glioblastoma, focusing on the markers Iba1 and CD163. The authors examine the different sources of TAMs, their activation, and their implications in glioma growth and treatment. The integration of artificial intelligence for macrophage identification in tissue samples is a particularly noteworthy advancement.
- The review could be enhanced by providing more detailed explanations of the immunomodulatory mechanisms by which Iba1 and CD163 exert their effects on gliomas
We have added the following new paragraphs after “role of iba1 in glioblastoma progression” at lines 262-272, 283-291.
Glioma-derived macrophage colony-stimulating factor (M-CSF) plays a crucial role in modulating the behavior of microglia and macrophages within the tumor microenvironment. Not only does M-CSF regulate Iba1 translocation and microglial motility, but it also acts as a potent inducer of angiogenesis through the upregulation of insulin-like growth factor-binding protein 1 (IGFBP1) (93). The importance of IGFBP1 in this process is underscored by the finding that silencing IGFBP1 results in a significant decrease in tubular formation in endothelial cells (93). Furthermore, IGFBP1 has been shown to promote GBM growth and survival by acting as an insulin-like growth factor, binding to prosurvival BAK, and attenuating the growth-inhibitory effects of p53 (94). Additionally, IGFBP1 is a potent activator of nitric oxide production through the phosphoinositide 3-kinase (PI3K) signaling pathway, which potentiates angiogenesis (94,95).
Our recent work suggests that there is a potential crosstalk between GSCs and Iba1+ TAMs (63). IL-33 has been shown to act as a bidirectional messenger between these cells: GSCs release IL-33, which binds to ST2 receptors on TAMs (98), increasing downstream STAT3 protein phosphorylation and upregulating IL-6 and LIF (99). This actively recruits TAMs, including Iba1+ TAMs (100), to the TME, and switches TAMs from an anti-tumorigenic state to a pro-tumorigenic, anti-inflammatory phenotype that promotes gliomagenesis. Interestingly, Iba1+ TAMs can also produce IL-33, which in turn is expected to feed back to GSCs, allowing them to maintain self-renewal and sustain stemness, supporting GBM progression (63,100).
References:
- Zheng Y, Fuse H, Alzoubi I, Graeber MB. Microglia-Derived Brain Macrophages Associate with Glioblastoma Stem Cells: A Potential Mechanism for Tumor Progression Revealed by AI-Assisted Analysis. Cells. 2025 Jan;14(6):413.
- Nijaguna MB, Patil V, Urbach S, Shwetha SD, Sravani K, Hegde AS, et al. Glioblastoma-derived Macrophage Colony-stimulating Factor (MCSF) Induces Microglial Release of Insulin-like Growth Factor-binding Protein 1 (IGFBP1) to Promote Angiogenesis. J Biol Chem. 2015 Sep 18;290(38):23401–15.
- Leu JIJ, George DL. Hepatic IGFBP1 is a prosurvival factor that binds to BAK, protects the liver from apoptosis, and antagonizes the proapoptotic actions of p53 at mitochondria. Genes Dev. 2007 Dec 1;21(23):3095–109.
- Rajwani A, Ezzat V, Smith J, Yuldasheva NY, Duncan ER, Gage M, et al. Increasing circulating IGFBP1 levels improves insulin sensitivity, promotes nitric oxide production, lowers blood pressure, and protects against atherosclerosis. Diabetes. 2012 Apr;61(4):915–24.
- Zhou Y, Xu Z, Liu Z. Role of IL-33-ST2 pathway in regulating inflammation: current evidence and future perspectives. J Transl Med. 2023 Dec 11;21(1):902.
- Johnson DE, O’Keefe RA, Grandis JR. Targeting the IL-6/JAK/STAT3 signalling axis in cancer. Nat Rev Clin Oncol. 2018 Apr;15(4):234–48.
- De Boeck A, Ahn BY, D’Mello C, Lun X, Menon SV, Alshehri MM, et al. Glioma-derived IL-33 orchestrates an inflammatory brain tumor microenvironment that accelerates glioma progression. Nat Commun. 2020 Oct 5;11(1):4997.
CD163 is discussed under “Regulation of CD163 expression and its implications for the immune response” and “Role of CD163 in antigen presentation and cytokine production”. In addition, the following paragraph have been added at lines 362-372:
CD163 is regarded as a marker of M2 macrophages (145). CD163+ TAMs secrete pleiotrophin (PTN), which binds to its receptor on GSCs, protein tyrosine phosphatase receptor type Z1 (PTPRZ1), promoting self-renewal and sustainability that support tumor progression (146). Like Iba1, CD163 expression can also be enhanced by increased IL-33 levels and its downstream IL-6/STAT3 signaling pathway, contributing to TAM recruitment and an anti-inflammatory shift (100). The secretion of IL-6, facilitated by hypoxic glioma-derived exosomes, has an additive effect on increasing CD163 and IL-10 expression via the same STAT3 pathway, thereby enhancing tumor progression (147). CD163+ TAMs have also been implicated in the release of C-C motif chemokine ligand 2 (CCL2), another essential cytokine for the recruitment of MDSCs and regulatory T cells (148).
References:
- De Boeck A, Ahn BY, D’Mello C, Lun X, Menon SV, Alshehri MM, et al. Glioma-derived IL-33 orchestrates an inflammatory brain tumor microenvironment that accelerates glioma progression. Nat Commun. 2020 Oct 5;11(1):4997.
- Barros MHM, Hauck F, Dreyer JH, Kempkes B, Niedobitek G. Macrophage Polarisation: an Immunohistochemical Approach for Identifying M1 and M2 Macrophages. PLOS ONE. 2013 Nov 15;8(11):e80908.
- Shi Y, Ping YF, Zhou W, He ZC, Chen C, Bian BSJ, et al. Tumour-associated macrophages secrete pleiotrophin to promote PTPRZ1 signalling in glioblastoma stem cells for tumour growth. Nat Commun. 2017 Jun 1;8:15080.
- Xu J, Zhang J, Zhang Z, Gao Z, Qi Y, Qiu W, et al. Hypoxic glioma-derived exosomes promote M2-like macrophage polarization by enhancing autophagy induction. Cell Death Dis. 2021 Apr 7;12(4):1–16.
- Chang AL, Miska J, Wainwright DA, Dey M, Rivetta CV, Yu D, et al. CCL2 Produced by the Glioma Microenvironment Is Essential for the Recruitment of Regulatory T Cells and Myeloid-Derived Suppressor Cells. Cancer Res. 2016 Oct 2;76(19):5671–82.
- Incorporate a dedicated section summarizing pertinent clinical data, particularly studies that assess the outcome of therapies targeting TAMs and their relationship with Iba1 and CD163 expression. This addition could enhance the clinical relevance of the manuscript significantly.
We refer to our reply to comment 4 of reviewer 1. The following information has been added at lines 647-667:
Therapeutic agents aimed at targeting TAMs in the TME to combat GBM progression have recently gained significant attention. M-CSF, a cytokine essential for TAM differentiation and GBM progression (210), has emerged as a promising target candidate for reducing GBM growth. Inhibiting M-CSF or its receptor, colony-stimulating factor-1 receptor (CSF-1R), has shown promise in preclinical GBM models (211). A small oral agent, PLX3397, was developed to directly target CSF-1R, demonstrating efficacy in GBM preclinical studies (212). However, human clinical trials (NCT01349036, NCT01790503) have re-ported limited efficacy (213,214). Another CSF-1R inhibitor, BLZ945, has been shown to improve survival rates (211) and enhance radiotherapy efficacy in glioma mouse models (215). In humans, BLZ945 can augment the efficacy of anti-PD-1 therapy against GBM by attenuating the immunosuppressive polarization of CD163+ macrophages (216). However, prolonged treatment with BLZ945 has yielded no additional benefits (217).
CD47, a "don't eat me" signal on tumor cells (218), has also been investigated as a potential target. An in vivo study found increased GBM phagocytosis following anti-CD47 treatment (219), whereas another model demonstrated minimal effects on glioma growth with anti-CD47 monotherapy (220). Moreover, anti-CD47 treatment can induce haematological toxicity (221). To date, human studies on anti-CD47 therapy are limited, and further research is needed to fully understand its potential. The TGF-βR1 inhibitor SB431542 has been shown to impair tumor growth by 75% in a mouse model (222), but the translatability of these findings to human studies remains uncertain (223,224).
References:
- Wu K, Lin K, Li X, Yuan X, Xu P, Ni P, et al. Redefining Tumor-Associated Macrophage Subpopulations and Func-tions in the Tumor Microenvironment. Front Immunol. 2020 Aug 4;11:1731.
- Pyonteck SM, Akkari L, Schuhmacher AJ, Bowman RL, Sevenich L, Quail DF, et al. CSF-1R inhibition alters mac-rophage polarization and blocks glioma progression. Nat Med. 2013 Oct;19(10):1264–72.
- Yan D, Kowal J, Akkari L, Schuhmacher AJ, Huse JT, West BL, et al. Inhibition of colony stimulating factor-1 recep-tor abrogates microenvironment-mediated therapeutic resistance in gliomas. Oncogene. 2017 Oct 26;36(43):6049–58.
- Butowski NA, Colman H, De Groot JF, Omuro AMP, Nayak L, Cloughesy TF, et al. A phase 2 study of orally ad-ministered PLX3397 in patients with recurrent glioblastoma. J Clin Oncol. 2014 May 20;32(15_suppl):2023–2023.
- Mendez JS, Cohen AL, Eckenstein M, Jensen RL, Burt LM, Salzman KL, et al. Phase 1b/2 study of orally adminis-tered pexidartinib in combination with radiation therapy and temozolomide in patients with newly diagnosed glioblastoma. Neuro-Oncol Adv. 2024 Nov 22;6(1):vdae202.
- ALMAHARIQ FM, QUINN JT, KESARWANI P, KANT S, MILLER CR, CHINNAIYAN P. Inhibition of Colony-Stimulating Factor-1 Receptor Enhances the Efficacy of Radiotherapy and Reduces Immune Suppression in Glio-blastoma. In Vivo. 2021 Jan 3;35(1):119–29.
- Cui X, Ma C, Vasudevaraja V, Serrano J, Tong J, Peng Y, et al. Dissecting the immunosuppressive tumor microenvi-ronments in Glioblastoma-on-a-Chip for optimized PD-1 immunotherapy. eLife. 9:e52253.
- Quail DF, Bowman RL, Akkari L, Quick ML, Schuhmacher AJ, Huse JT, et al. The tumor microenvironment under-lies acquired resistance to CSF1R inhibition in gliomas. Science. 2016 May 20;352(6288):aad3018.
- Willingham SB, Volkmer JP, Gentles AJ, Sahoo D, Dalerba P, Mitra SS, et al. The CD47-signal regulatory protein alpha (SIRPa) interaction is a therapeutic target for human solid tumors. Proc Natl Acad Sci U S A. 2012 Apr 24;109(17):6662–7.
- Zhang M, Hutter G, Kahn SA, Azad TD, Gholamin S, Xu CY, et al. Anti-CD47 Treatment Stimulates Phagocytosis of Glioblastoma by M1 and M2 Polarized Macrophages and Promotes M1 Polarized Macrophages In Vivo. PLOS ONE. 2016 Apr 19;11(4):e0153550.
- von Roemeling CA, Wang Y, Qie Y, Yuan H, Zhao H, Liu X, et al. Therapeutic modulation of phagocytosis in glio-blastoma can activate both innate and adaptive antitumour immunity. Nat Commun. 2020 Mar 20;11(1):1508.
- Sikic BI, Lakhani N, Patnaik A, Shah SA, Chandana SR, Rasco D, et al. First-in-Human, First-in-Class Phase I Trial of the Anti-CD47 Antibody Hu5F9-G4 in Patients With Advanced Cancers. J Clin Oncol. 2019 Apr 20;37(12):946–53.
- Liu H, Sun Y, Zhang Q, Jin W, Gordon RE, Zhang Y, et al. Pro-inflammatory and proliferative microglia drive pro-gression of glioblastoma. Cell Rep [Internet]. 2021 Sep 14 [cited 2025 Apr 17];36(11). Available from: https://www.cell.com/cell-reports/abstract/S2211-1247(21)01167-0
- Wick A, Desjardins A, Suarez C, Forsyth P, Gueorguieva I, Burkholder T, et al. Phase 1b/2a study of galunisertib, a small molecule inhibitor of transforming growth factor-beta receptor I, in combination with standard te-mozolomide-based radiochemotherapy in patients with newly diagnosed malignant glioma. Invest New Drugs. 2020;38(5):1570–9.
- Hollander MW den, Bensch F, Glaudemans AWJM, Munnink THO, Enting RH, Dunnen WFA den, et al. TGF-β Antibody Uptake in Recurrent High-Grade Glioma Imaged with 89Zr-Fresolimumab PET. J Nucl Med. 2015 Sep 1;56(9):1310–4.
- The association of Iba1 with signaling pathways such as TGF-β is particularly insightful. The review highlights how TGF-β promotes immunosuppression and contributes to gliomagenesis, thereby implicating Iba1's involvement in critical tumor progression mechanisms. However, there could be a more thorough exploration of how Iba1 interacts with other signaling pathways and its direct effects on tumor cell behavior and immune evasion
We refer to our reply to comment 1 regarding Iba1.
- The manuscript should emphasize the therapeutic implications of targeting Iba1. Given its role in glioblastoma progression, there is a clear opportunity for developing therapies that modulate Iba1’s activity or its associated pathways. The discussion of preclinical data supporting this concept is valuable, but further elaboration on potential therapeutic modalities or ongoing clinical trials would strengthen this section
To date, no preclinical or clinical studies have been conducted on targeting Iba1 as a therapeutic strategy.
- The review mentions Iba1’s involvement in angiogenesis through its association with pro-angiogenic factors. Expanding this discussion could provide deeper insights into how increased Iba1 expression facilitates neovascularization within tumors.
The following original text in section “Role of Iba1 in glioblastoma progression” at lines 250-261 discussing the association between Iba1 and pro-angiogenic factors.
“The translocation of FOXC2 in human brain microvessel endothelial cells (hBMECs) has been shown to upregulate angiogenesis through both direct and indirect mechanisms. Furthermore, TGF-β has been found to be directly linked to FOXC2 via the EMT, suggesting an interplay between these factors in promoting angiogenic processes (91). Having a strong correlation with TAM activation, GBM-associated polymorphonuclear leukocytes/granulocytes (GBM-hPMNL) have been identified, where the levels of pro-angiogenic cytokines and factors such as CXC motif chemokine ligand 2 (CXCL2), TEK, CD163, and Hypoxia inducible factor-1α (HIF-1α) are increased (92). These pro-angiogenic granulocytes were shown to facilitate remodelling and regeneration of tumoral blood vessels in GBM (92). A significant increase of CXCL2, a member of the CXC family, was demonstrated by Brandenburg et al. within GBM-bearing mouse brain, and its angiogenic effect was found to be even stronger than that of VEGF (79).”
The following text is added to expand the involvement of Iba1 in angiogenesis at lines 262-272:
Glioma-derived macrophage colony-stimulating factor (M-CSF) plays a crucial role in modulating the behavior of microglia and macrophages within the tumor microenvironment. Not only does M-CSF regulate Iba1 translocation and microglial motility, but it also acts as a potent inducer of angiogenesis through the upregulation of insulin-like growth factor-binding protein 1 (IGFBP1) (93). The importance of IGFBP1 in this process is underscored by the finding that silencing IGFBP1 results in a significant decrease in tubular formation in endothelial cells (93). Furthermore, IGFBP1 has been shown to promote GBM growth and survival by acting as an insulin-like growth factor, binding to prosurvival BAK, and attenuating the growth-inhibitory effects of p53 (94). Additionally, IGFBP1 is a potent activator of nitric oxide production through the phosphoinositide 3-kinase (PI3K) signaling pathway, which potentiates angiogenesis (94,95)
References:
- Nijaguna MB, Patil V, Urbach S, Shwetha SD, Sravani K, Hegde AS, et al. Glioblastoma-derived Macrophage Colo-ny-stimulating Factor (MCSF) Induces Microglial Release of Insulin-like Growth Factor-binding Protein 1 (IGFBP1) to Promote Angiogenesis. J Biol Chem. 2015 Sep 18;290(38):23401–15.
- Leu JIJ, George DL. Hepatic IGFBP1 is a prosurvival factor that binds to BAK, protects the liver from apoptosis, and antagonizes the proapoptotic actions of p53 at mitochondria. Genes Dev. 2007 Dec 1;21(23):3095–109.
- Rajwani A, Ezzat V, Smith J, Yuldasheva NY, Duncan ER, Gage M, et al. Increasing circulating IGFBP1 levels im-proves insulin sensitivity, promotes nitric oxide production, lowers blood pressure, and protects against athero-sclerosis. Diabetes. 2012 Apr;61(4):915–24.
- The discussion surrounding CD163's role in antigen presentation and its regulation in response to pro-inflammatory and anti-inflammatory signals is crucial. CD163, as a marker for TAMs, is pivotal in understanding the immune landscape of the tumor microenvironment in glioblastoma, particularly how these cells may facilitate or hinder anti-tumor immunity
We refer to our reply to comment 1 regarding CD163.
- A list of abbreviations should be added to the manuscript
The requested list of abbreviations has been added to the manuscript.
Reviewer 5 Report
Comments and Suggestions for Authors
This review article focuses on the roles of Iba1 and CD163 in malignant gliomas, particularly glioblastoma. It discusses the characteristics of tumor-associated macrophages (TAMs) in the tumor microenvironment (TME) of gliomas and their involvement in glioma progression. The review also examines the expression, functions, and regulation of Iba1 and CD163 in both normal and pathological brain tissue, and explores their potential as therapeutic targets. Additionally, the application of AI-enhanced techniques in analyzing TAMs is highlighted.
Comments:
Abstract, line 34: "unique and shared roles" change to "unique and common roles"
Update the statistics on overall cancer incidence and the prevalence of this specific cancer type, including survival rates, to emphasize the urgent need for cancer studies. Cite Cancer Statistics, 2024. Additionally, provide a general overview of cancer therapy, referencing the NIH paper“Cancer treatments: Past, present, and future, 2024” for further insights.When introduce glioma, mention some recently discovered glioma biomarkers and cite related studies, such as MAPK signaling pathway‑based glioma subtypes, glioma SCN3B and glioma CDK2. The increasingly appreciated role of tumor-associated microglia/macrophages (TAMs) in glioblastoma, line 45: "tumor-associated microglia/macrophages" change to "tumor-associated microglia / macrophages"
The increasingly appreciated role of tumor-associated microglia/macrophages (TAMs) in glioblastoma, line 51: needs to cite more reference to support "TAMs can be derived from resident microglia, perivascular and infiltrating blood-derived macrophages"
Overview of Ibal structure, function and its expression in normal and pathologically altered brain tissue, line 82: "white and grey matters" change to "white and gray matter"
Figure 1 caption, line 99: needs to cite more reference to support "gradient of microglial activation in areas of diffuse tumor infiltration"
Role of Ibal in glioblastoma progression, line 114: needs to cite more reference to support "How Ibal+ microglia contribute to the progression of GBM is of great interest and under intense scrutiny."
Figure 2 caption, line 157: needs to cite more reference to support the whole caption.
Ibal as a potential future therapeutic target in glioblastoma, line 209: "three-pronged strategy" change to "three-step strategy"
Table 1, add more references in the whole table.
CD163, a macrophage scavenger protein showing differential expression in glioblastoma, line 239: needs to cite more reference to support "Perivascular macrophages (PVMs) are a special population of macrophages reside within the perivascular spaces"
Figure 3 caption, line 343, and Figure 4 caption, line 354: needs to cite more reference to support the whole captions.
Role of microglia and macrophages in glioblastoma progression, line 477: "mutated transcript that allows longer survival(2)" change to "mutated transcript that allows longer survival " Suggest future studies that could validate these findings in patient-derived xenograft models. Previous studies using xenograft models of cancer should be mentioned, such as “Comparing volatile and intravenous anesthetics in a mouse model of breast cancer metastasis, 2018,Deficiency of BCAT2-mediated branched-chain amino acid catabolism promotes colorectal cancer development, 2024,hsa-miR-CHA2, a novel microRNA, exhibits anticancer effects by suppressing cyclin E1 in human non-small cell lung cancer cells, 2024”
Recent studies have highlighted advancements in liquid biopsies for cancer diagnostics and monitoring. Research such as “Updates on liquid biopsies in neuroblastoma for treatment response, relapse and recurrence assessment, 2024”demonstrates the utility of circulating tumor DNA (ctDNA) detection through liquid biopsy techniques. Additionally, emerging sequencing technologies have improved the sensitivity and specificity of DNA analysis, such as “Development of a molecular barcode detection system for pancreaticobiliary malignancies and comparison with next-generation sequencing, 2024”. Also the methylation is also used for detection, reported in “Methylation signatures as biomarkers for non-invasive early detection of breast cancer: A systematic review of the literature, 2024”. Please cited these related papers and discuss: consider whether the mechanisms discussed in this study could be identified through these diagnosis methods.
Conclusion, line 528: "The binary M1/M2 classification has become unpopular" change to "The binary M1/M2 classification has become less popular"
Author Response
This review article focuses on the roles of Iba1 and CD163 in malignant gliomas, particularly glioblastoma. It discusses the characteristics of tumor-associated macrophages (TAMs) in the tumor microenvironment (TME) of gliomas and their involvement in glioma progression. The review also examines the expression, functions, and regulation of Iba1 and CD163 in both normal and pathological brain tissue, and explores their potential as therapeutic targets. Additionally, the application of AI-enhanced techniques in analyzing TAMs is highlighted.
Comments:
- Abstract, line 34: "unique and shared roles" change to "unique and common roles"
The change has been made.
- Update the statistics on overall cancer incidence and the prevalence of this specific cancer type, including survival rates, to emphasize the urgent need for cancer studies. Cite Cancer Statistics, 2024. Additionally, provide a general overview of cancer therapy, referencing the NIH paper“Cancer treatments: Past, present, and future, 2024” for further insights.When introduce glioma, mention some recently discovered glioma biomarkers and cite related studies, such as MAPK signaling pathway‑based glioma subtypes, glioma SCN3B and glioma CDK2. The increasingly appreciated role of tumor-associated microglia/macrophages (TAMs) in glioblastoma, line 45: "tumor-associated microglia/macrophages" change to "tumor-associated microglia / macrophages"
This comment is addressed in lines 45-65:
In 2024, Siegel et al. (1) estimated that 18,760 deaths from brain and other nervous system cancers would occur in the United States alone in that year. Glioblastoma (GBM) accounts for nearly half of all malignant brain tumors, with a median survival of just 14.6 months (2). Furthermore, the mortality rate for GBM has remained largely unchanged since 1975 (1), highlighting the persistent lack of significant therapeutic advances despite decades of medical research. In recent years, the emergence of targeted therapies — including immune checkpoint inhibitors and chimeric antigen receptor T-cell (CAR-T) therapy — has markedly improved outcomes in several cancer types (3). However, such innovative approaches have shown limited efficacy in GBM so far, and the cur-rent gold standard for treatment still relies on maximal safe surgical resection followed by concurrent radiotherapy and temozolomide chemotherapy (4).
Molecular profiling studies are providing new insights into potential therapies. For instance, a study published by Liu and Tang (5) reported the discovery of 127 mitogen-activated protein kinase (MAPK) genes upregulated in glioma, allowing for a potential C1/C2 classification of glioma subtypes with distinct prognostic implications. Similarly, voltage-gated sodium channel β3 subunit (SCN3B) has been identified as a prognostic biomarker for gliomas, specifically oligodendroglioma (6), where higher levels of SCN3B correlate with longer survival. CDK2, a key cell cycle regulator, is associated with GBM prognosis. Furthermore, lower CDK2 levels indicate a higher response to immunotherapy (7), suggesting its potential role as a biomarker and a therapeutic target. These findings suggest new avenues for personalized and targeted therapies in GBM.
References:
- Siegel RL, Giaquinto AN, Jemal A. Cancer statistics, 2024. CA Cancer J Clin. 2024;74(1):12–49.
- Grochans S, Cybulska AM, SimiÅ„ska D, Korbecki J, Kojder K, Chlubek D, et al. Epidemiology of Glioblastoma Multiforme–Literature Review. Cancers. 2022 May 13;14(10):2412.
- Sonkin D, Thomas A, Teicher BA. Cancer treatments: Past, present, and future. Cancer Genet. 2024 Aug;286–287:18–24.
- Kotecha R, Odia Y, Khosla AA, Ahluwalia MS. Key Clinical Principles in the Management of Glioblastoma. JCO Oncol Pract. 2023 Apr;19(4):180–9.
- Liu H, Tang T. MAPK signaling pathway-based glioma subtypes, machine-learning risk model, and key hub proteins identification. Sci Rep. 2023 Nov 4;13(1):19055.
- Liu H, Weng J, Huang CLH, Jackson AP. Is the voltage-gated sodium channel β3 subunit (SCN3B) a biomarker for glioma? Funct Integr Genomics. 2024 Sep 18;24(5):162.
- Liu H, Weng J. A comprehensive bioinformatic analysis of cyclin-dependent kinase 2 (CDK2) in glioma. Gene. 2022 May 15;822:146325.
The change in line 45 change has been made. - The increasingly appreciated role of tumor-associated microglia/macrophages (TAMs) in glioblastoma, line 51: needs to cite more reference to support "TAMs can be derived from resident microglia, perivascular and infiltrating blood-derived macrophages"
The following references have been added to the respective sentences.
Buonfiglioli, A., & Hambardzumyan, D. (2021). Macrophages and microglia: the cerberus of glioblastoma. Acta neuropathologica communications, 9(1), 54. https://doi.org/10.1186/s40478-021-01156-z
Hambardzumyan, D., Gutmann, D. & Kettenmann, H. The role of microglia and macrophages in glioma maintenance and progression. Nat Neurosci 19, 20–27 (2016). https://doi.org/10.1038/nn.4185
- Overview of Ibal structure, function and its expression in normal and pathologically altered brain tissue, line 82: "white and grey matters" change to "white and gray matter"
This change has been made.
- Figure 1 caption, line 99: needs to cite more reference to support "gradient of microglial activation in areas of diffuse tumor infiltration"
The following references have been added:
Zheng, Y., Fuse, H., Alzoubi, I., & Graeber, M. B. (2025). Microglia-Derived Brain Macrophages Associate with Glioblastoma Stem Cells: A Potential Mechanism for Tumor Progression Revealed by AI-Assisted Analysis. Cells, 14(6), 413. https://doi.org/10.3390/cells14060413
Loh, C., Zheng, Y., Alzoubi, I., Alexander, K. L., Lee, M., Cai, W. D., Song, Y., McDonald, K., Nowak, A. K., Banati, R. B., & Graeber, M. B. (2025). Microglia and brain macrophages are differentially associated with tumor necrosis in glioblastoma: A link to tumor progression. Oncology research, 33(4), 937–950. https://doi.org/10.32604/or.2024.056436
- Role of Ibal in glioblastoma progression, line 114: needs to cite more reference to support "How Ibal+ microglia contribute to the progression of GBM is of great interest and under intense scrutiny."
A distinction between Iba1+ microglia and Iba1+ macrophages should be made. The following references, relevant to Iba1+ microglia, have been added:
Zheng, Y., Fuse, H., Alzoubi, I., & Graeber, M. B. (2025). Microglia-Derived Brain Macrophages Associate with Glioblastoma Stem Cells: A Potential Mechanism for Tumor Progression Revealed by AI-Assisted Analysis. Cells, 14(6), 413. https://doi.org/10.3390/cells14060413
Zheng, Y., & Graeber, M. B. (2022). Microglia and Brain Macrophages as Drivers of Glioma Progression. International Journal of Molecular Sciences, 23(24), 15612. https://doi.org/10.3390/ijms232415612
De Boeck, A., Ahn, B. Y., D'Mello, C., Lun, X., Menon, S. V., Alshehri, M. M., Szulzewsky, F., Shen, Y., Khan, L., Dang, N. H., Reichardt, E., Goring, K. A., King, J., Grisdale, C. J., Grinshtein, N., Hambardzumyan, D., Reilly, K. M., Blough, M. D., Cairncross, J. G., Yong, V. W., … Senger, D. L. (2020). Glioma-derived IL-33 orchestrates an inflammatory brain tumor microenvironment that accelerates glioma progression. Nature communications, 11(1), 4997. https://doi.org/10.1038/s41467-020-18569-4
- Figure 2 caption, line 157: needs to cite more reference to support the whole caption
The following references have been added to the legend of Figure 2:
TGF-b:Hata, A., & Chen, Y. G. (2016). TGF-β Signaling from Receptors to Smads. Cold Spring Harbor perspectives in biology, 8(9), a022061. https://doi.org/10.1101/cshperspect.a022061
Derynck, R., Zhang, Y. Smad-dependent and Smad-independent pathways in TGF-β family signalling. Nature 425, 577–584 (2003). https://doi.org/10.1038/nature02006
TSP1: Joseph, J. V., Magaut, C. R., Storevik, S., Geraldo, L. H., Mathivet, T., Latif, M. A., Rudewicz, J., Guyon, J., Gambaretti, M., Haukas, F., Trones, A., Rømo Ystaas, L. A., Hossain, J. A., Ninzima, S., Cuvellier, S., Zhou, W., Tomar, T., Klink, B., Rane, L., Irving, B. K., … Miletic, H. (2022). TGF-β promotes microtube formation in glioblastoma through thrombospondin 1. Neuro-oncology, 24(4), 541–553. https://doi.org/10.1093/neuonc/noab212
Bikfalvi, A., Guyon, J., & Daubon, T. (2024). New insights into the role of thrombospondin-1 in glioblastoma development. Seminars in cell & developmental biology, 155(Pt B), 52–57. https://doi.org/10.1016/j.semcdb.2023.09.001
FOXC2: Sano, H., Leboeuf, J. P., Novitskiy, S. V., Seo, S., Zaja-Milatovic, S., Dikov, M. M., & Kume, T. (2010). The Foxc2 transcription factor regulates tumor angiogenesis. Biochemical and biophysical research communications, 392(2), 201–206. https://doi.org/10.1016/j.bbrc.2010.01.015
- Ibal as a potential future therapeutic target in glioblastoma, line 209: "three-pronged strategy" change to "three-step strategy"
This change has been made.
- Table 1, add more references in the whole table.
The following references have been added:
Iba1 as a biomarker:
Liu X, Zhang D, Hu J, Xu S, Xu C, Shen Y. Allograft inflammatory factor 1 is a potential diagnostic, immunological, and prognostic biomarker in pan-cancer. Aging. 2023 Apr 3;15(7):2582.
Molecular mechanisms:
IL-10: Ip, W. K. E., Hoshi, N., Shouval, D. S., Snapper, S., & Medzhitov, R. (2017). Anti-inflammatory effect of IL-10 mediated by metabolic reprogramming of macrophages. Science (New York, N.Y.), 356(6337), 513–519. https://doi.org/10.1126/science.aal3535
Palmieri, E. M., Menga, A., Martín-Pérez, R., Quinto, A., Riera-Domingo, C., De Tullio, G., Hooper, D. C., Lamers, W. H., Ghesquière, B., McVicar, D. W., Guarini, A., Mazzone, M., & Castegna, A. (2017). Pharmacologic or Genetic Targeting of Glutamine Synthetase Skews Macrophages toward an M1-like Phenotype and Inhibits Tumor Metastasis. Cell reports, 20(7), 1654–1666. https://doi.org/10.1016/j.celrep.2017.07.054
HIF-1a: Dietl, K., Renner, K., Dettmer, K., Timischl, B., Eberhart, K., Dorn, C., Hellerbrand, C., Kastenberger, M., Kunz-Schughart, L. A., Oefner, P. J., Andreesen, R., Gottfried, E., & Kreutz, M. P. (2010). Lactic acid and acidification inhibit TNF secretion and glycolysis of human monocytes. Journal of immunology (Baltimore, Md. : 1950), 184(3), 1200–1209. https://doi.org/10.4049/jimmunol.0902584
Wang, T., Liu, H., Lian, G., Zhang, S. Y., Wang, X., & Jiang, C. (2017). HIF1α-Induced Glycolysis Metabolism Is Essential to the Activation of Inflammatory Macrophages. Mediators of inflammation, 2017, 9029327. https://doi.org/10.1155/2017/9029327
Tumour microenvironment:
(TGF-beta): Bedolla A, Wegman E, Weed M, Stevens MK, Ware K, Paranjpe A, et al. Adult microglial TGFβ1 is required for microglia homeostasis via an autocrine mechanism to maintain cognitive function in mice. Nat Commun. 2024 Jun 21;15(1):5306.
Ras/Raf/MEK/ERK: Zhang Y. E. (2009). Non-Smad pathways in TGF-beta signaling. Cell research, 19(1), 128–139. https://doi.org/10.1038/cr.2008.328
PI3k/AKT/mTOR: Lamouille S, Derynck R . Cell size and invasion in TGF-beta-induced epithelial to mesenchymal transition is regulated by activation of the mTOR pathway. J Cell Biol 2007; 178: 437–451.
FOXC2: Sano, H., Leboeuf, J. P., Novitskiy, S. V., Seo, S., Zaja-Milatovic, S., Dikov, M. M., & Kume, T. (2010). The Foxc2 transcription factor regulates tumor angiogenesis. Biochemical and biophysical research communications, 392(2), 201–206. https://doi.org/10.1016/j.bbrc.2010.01.015
- CD163, a macrophage scavenger protein showing differential expression in glioblastoma, line 239: needs to cite more reference to support "Perivascular macrophages (PVMs) are a special population of macrophages reside within the perivascular spaces"
The following references have been added:
Zheng L, Guo Y, Zhai X, Zhang Y, Chen W, Zhu Z, et al. Perivascular macrophages in the CNS: From health to neurovascular diseases. CNS Neurosci Ther. 2022 Sep 20;28(12):1908–20.
Wenjie Wen, Jinping Cheng, Yamei Tang, Brain perivascular macrophages: current understanding and future prospects, Brain, Volume 147, Issue 1, January 2024, Pages 39–55, https://doi.org/10.1093/brain/awad304
- Figure 3 caption, line 343, and Figure 4 caption, line 354: needs to cite more reference to support the whole captions.
The following text has been added to the legend of both figures:
For more information on the labelling used please refer to (70).
- Loh C, Zheng Y, Alzoubi I, Alexander KL, Lee M, Cai WD, et al. Microglia and brain macrophages are differentially associated with tumor necrosis in glioblastoma: A link to tumor progression. Oncol Res. 2024;0(0):1–10.
- Role of microglia and macrophages in glioblastoma progression, line 477: "mutated transcript that allows longer survival(2)" change to "mutated transcript that allows longer survival "
This change has been made
14: Suggest future studies that could validate these findings in patient-derived xenograft models. Previous studies using xenograft models of cancer should be mentioned, such as “Comparing volatile and intravenous anesthetics in a mouse model of breast cancer metastasis, 2018, Deficiency of BCAT2-mediated branched-chain amino acid catabolism promotes colorectal cancer development, 2024,hsa-miR-CHA2, a novel microRNA, exhibits anticancer effects by suppressing cyclin E1 in human non-small cell lung cancer cells, 2024”
The mentioned studies were cited and included at lines 631-645:
Patient-derived xenografts offer unique insights into cancer pathobiology and therapeutic target discovery. Recently, a mouse model (205) was used to study how different types of anaesthetics (volatile vs intravenous) affect breast cancer metastasis. This model successfully replicated the effects of anesthetic agents, shedding light on their underlying mechanisms. Similarly, other xenograft models, such as those established with colorectal cancer (206) and non-small cell lung cancer cells (207), have yielded valuable information on human cancer biology in vivo. In the context of GBM, several xenograft models have been developed, including heterotopic (implanted outside of the CNS) and orthotopic (implanted within the CNS) implantation approaches, as reviewed by Gómez-Oliva and colleagues (208). While heterotopic models facilitate direct observation of GBM development and provide a platform for preclinical drug testing, they fail to accurately recapitulate the microenvironment of the CNS (208). Consequently, orthotopic xenografts have gained popularity, as they more closely mimic actual GBM conditions and interactions with immune cells within the CNS, despite presenting technical challenges (209).
References
- Li R, Huang Y, Liu H, Dilger JP, Lin J. Abstract 2162: Comparing volatile and intravenous anesthetics in a mouse model of breast cancer metastasis. Cancer Res. 2018 Jul 1;78(13_Supplement):2162.
- Kang ZR, Jiang S, Han JX, Gao Y, Xie Y, Chen J, et al. Deficiency of BCAT2-mediated branched-chain amino acid catabolism promotes colorectal cancer development. Biochim Biophys Acta Mol Basis Dis. 2024 Feb;1870(2):166941.
- Lee SJ, Jeon SH, Cho S, Kim CM, Yoo JK, Oh SH, et al. hsa-miR-CHA2, a novel microRNA, exhibits anticancer effects by suppressing cyclin E1 in human non-small cell lung cancer cells. Biochim Biophys Acta Mol Basis Dis. 2024 Aug;1870(6):167250.
- Gómez-Oliva R, Domínguez-García S, Carrascal L, Abalos-Martínez J, Pardillo-Díaz R, Verástegui C, et al. Evolution of Experimental Models in the Study of Glioblastoma: Toward Finding Efficient Treatments. Front Oncol [Internet]. 2021 Jan 29 [cited 2025 Apr 17];10. Available from: https://www.frontiersin.orghttps://www.frontiersin.org/journals/oncology/articles/10.3389/fonc.2020.614295/full
- Shu Q, Wong KK, Su JM, Adesina AM, Yu LT, Tsang YTM, et al. Direct orthotopic transplantation of fresh surgical specimen preserves CD133+ tumor cells in clinically relevant mouse models of medulloblastoma and glioma. Stem Cells Dayt Ohio. 2008 Jun;26(6):1414–24.
15: Recent studies have highlighted advancements in liquid biopsies for cancer diagnostics and monitoring. Research such as “Updates on liquid biopsies in neuroblastoma for treatment response, relapse and recurrence assessment, 2024”demonstrates the utility of circulating tumor DNA (ctDNA) detection through liquid biopsy techniques. Additionally, emerging sequencing technologies have improved the sensitivity and specificity of DNA analysis, such as “Development of a molecular barcode detection system for pancreaticobiliary malignancies and comparison with next-generation sequencing, 2024”. Also the methylation is also used for detection, reported in “Methylation signatures as biomarkers for non-invasive early detection of breast cancer: A systematic review of the literature, 2024”. Please cited these related papers and discuss: consider whether the mechanisms discussed in this study could be identified through these diagnosis methods.
The above has been addressed at lines 599-630:
Recent studies have provided promising evidence for the usefulness of liquid biopsy in retrieving circulating tumor DNA (ctDNA), a type of DNA that is tumor-specific (194). The detection of ctDNA has been shown to greatly assist in the diagnosis of solid tumors, such as colorectal, breast, and lung cancers (195). However, the BBB poses a significant challenge in the case of GBM, with only approximately 1% of GBM ctDNA detectable in plasma (196). An alternative approach to overcome this limitation involves collecting samples from CSF, which may offer improved detection rates (195,197). Despite its potential as a diagnostic tool, liquid biopsy has limitations in providing insights into the dynamic nature of the TME in GBM, particularly with regards to real-time visualization of immune cell infiltration (198).
Molecular barcoding (MB) is an emerging next-generation sequencing technology that enhances accuracy and sensitivity in detecting rare DNA variants, offering significant improvements over traditional methods (199). Through the use of amplification techniques, MB enables the high-throughput identification of mutations from ctDNA, even when present at very low concentrations in plasma (199). However, despite these advancements, MB analysis of ctDNA has inherent limitations, notably its inability to comprehensively reveal the complex immune compositions of the TME. Fortunately, liquid biopsy approaches utilizing exosomal circular RNAs (circRNAs) have shown promise in addressing this challenge. circRNAs are particularly valuable as they reflect, in real-time, how glioma cells and immune cells respond to environmental changes, thereby providing a dynamic snapshot of the TME (200).
Furthermore, the methylation signature has emerged as an innovative tool for assisting in the early detection of malignancy (201). Methylation signature refers to the specific pattern of DNA methylation, characterized by the addition of a methyl group to cytosine residues, which yields a unique epigenetic fingerprint for a particular disease or cell type (202). Recent research has identified distinct methylation signatures associated with long-term survivors of GBM, defined as patients who exhibit extended survival beyond typical expectations after diagnosis (203). These findings suggest that methylation signature analysis may offer valuable insights into the biological characteristics distinguishing this subgroup of patients. Moreover, this technology holds promise for investigating the origins of TAMs, characterizing their diverse states, and detecting epigenetic changes in TAMs induced by glioma cells (204).
References:
- Jahangiri L. Updates on liquid biopsies in neuroblastoma for treatment response, relapse and recurrence assess-ment. Cancer Genet. 2024 Nov;288–289:32–9.
- Otsuji R, Fujioka Y, Hata N, Kuga D, Hatae R, Sangatsuda Y, et al. Liquid Biopsy for Glioma Using Cell-Free DNA in Cerebrospinal Fluid. Cancers. 2024 Jan;16(5):1009.
- Piccioni DE, Achrol ,Achal Singh, Kiedrowski ,Lesli A, Banks ,Kimberly C, Boucher ,Najee, Barkhoudarian ,Garni, et al. Analysis of cell-free circulating tumor DNA in 419 patients with glioblastoma and other primary brain tu-mors. CNS Oncol. 2019 Jun 30;8(2):CNS34.
- Pan W, Gu W, Nagpal S, Gephart MH, Quake SR. Brain Tumor Mutations Detected in Cerebral Spinal Fluid. Clin Chem. 2015 Mar 1;61(3):514–22.
- Müller Bark J, Kulasinghe A, Chua B, Day BW, Punyadeera C. Circulating biomarkers in patients with glioblasto-ma. Br J Cancer. 2020 Feb;122(3):295–305.
- Ohyama H, Hirotsu Y, Amemiya K, Mikata R, Amano H, Hirose S, et al. Development of a molecular barcode de-tection system for pancreaticobiliary malignancies and comparison with next-generation sequencing. Cancer Genet. 2024 Jan;280–281:6–12.
- Wu X, Shi M, Lian Y, Zhang H. Exosomal circRNAs as promising liquid biopsy biomarkers for glioma. Front Im-munol. 2023 Apr 14;14:1039084.
- Gonzalez T, Nie Q, Chaudhary LN, Basel D, Reddi HV. Methylation signatures as biomarkers for non-invasive early detection of breast cancer: A systematic review of the literature. Cancer Genet. 2024 Apr;282–283:1–8.
- Li Y, Fan Z, Meng Y, Liu S, Zhan H. Blood-based DNA methylation signatures in cancer: A systematic review. Bio-chim Biophys Acta Mol Basis Dis. 2023 Jan 1;1869(1):166583.
- Decraene B, Coppens G, Spans L, Solie L, Sciot R, Vanden Bempt I, et al. A novel methylation signature predicts extreme long-term survival in glioblastoma. J Neurooncol. 2024;169(2):341–7.
- Torrisi F, D’Aprile S, Denaro S, Pavone AM, Alberghina C, Zappalà A, et al. Epigenetics and Metabolism Repro-gramming Interplay into Glioblastoma: Novel Insights on Immunosuppressive Mechanisms. Antioxidants. 2023 Jan 18;12(2):220.
16: Conclusion, line 528: "The binary M1/M2 classification has become unpopular" change to "The binary M1/M2 classification has become less popular"
This change has been made.
Round 2
Reviewer 2 Report
Comments and Suggestions for Authors
The authors have addressed all the comments, the manuscript can be accepted for publication.
Reviewer 3 Report
Comments and Suggestions for Authors
My comments have been taken into account, the manuscript can be accepted.
Reviewer 5 Report
Comments and Suggestions for Authors
ok
Comments on the Quality of English Languageok